# OCD: Learning to Overfit with Conditional Diffusion Models

## Abstract

We present a dynamic model in which the weights are conditioned on an input sample $x$ and are learned to match those that would be obtained by finetuning a base model on $x$ and its label $y$. This mapping between an input sample and network weights is shown to be approximated by a linear transformation of the sample distribution, which suggests that a denoising diffusion model can be suitable for this task. The diffusion model we therefore employ focuses on modifying a single layer of the base model and is conditioned on the input, activations, and output of this layer. Our experiments demonstrate the wide applicability of the method for image classification, 3D reconstruction, tabular data, speech separation, and natural language processing. Our code is attached as supplementary.

## 1 Introduction

> Here is a simple local algorithm: For each testing pattern, (1) select the few training examples located in the vicinity of the testing pattern, (2) train a neural network with only these few examples, and (3) apply the resulting network to the testing pattern.
>
> Bottou & Vapnik (1992)

Thirty years after the local learning method in the epigraph was introduced, it can be modernized in a few ways. First, instead of training a neural network from scratch on a handful of samples, the method can finetune, with the same samples, a base model that is pretrained on the entire training set. The empirical success of transfer learning methods (Han et al., 2021) suggests that this would lead to an improvement.

Second, instead of retraining a neural network each time, we can learn to predict the weights of the locally-trained neural network for each input sample. This idea utilizes a dynamic, input-dependent architecture, also known as a hypernetwork (Ha et al., 2016).

Third, we can take the approach to an extreme and consider local regions that contain a single sample. During training, we finetune the base model for each training sample separately. In this process, which we call "overfitting", we train on each specific sample $s = (x, y)$ from the training set, starting with the weights of the base model and obtaining a model $f_{\theta_s}$. We then learn a model $g$ that maps between $x$ (without the label) and the shift in the weights of $f_{\theta_s}$ from those of the base model. Given a test sample $x$, we apply the learned mapping $g$ to it, obtain model weights, and apply the resulting model to $x$.

The overfitted models are expected to be similar to the base model, since the samples we overfit are part of the training set of the base model. We provide theoretical arguments that support that the mapping from the $x$ part of $s$ to $f_{\theta_s}$ can be approximated by a locally convex transformation. As a result, it is likely that a diffusion process that is able to generate samples in the domain of $x$ would also work for generating the weights of the fine-tuned networks. Recently, diffusion models, such as DDPM (Ho et al., 2020) and DDIM (Song et al., 2020) were shown to be highly successful in generating perceptual samples (Dhariwal & Nichol, 2021b; Kong et al., 2021). We, therefore, employ a conditional diffusion model to model $g$.

In order to make the diffusion models suitable for predicting network weights, we make three adjustments. First, we automatically select a specific layer of the neural model and modify only this

layer. This considerably reduces the size of the generated data and, in our experience, is sufficient for supporting the overfitting effect. Second, we condition the diffusion process on the input of the selected layer, its activations, and its output. Third, since the diffusion process assumes unit variance scale (Ho et al., 2020), we separately learn the scale of the weight modification.

Our method is widely applicable, and we evaluate it across four very different domains: image classification, image synthesis, regression in tabular data, and speech separation. In all cases, the results obtained by our method improve upon the non-local use of the same underlying architecture.

## 2 RELATED WORK

**Local learning** approaches perform inference with models that are focused on training samples in the vicinity of each test sample. This way, the predictions are based on what is believed to be the most relevant data points. K-nearest neighbors, for example, is a local learning method. Bottou & Vapnik (1992) have presented a simple algorithm for adjusting the capacity of the learned model locally, and discuss the advantages of such models for learning with uneven data distributions. Alpaydin & Jordan (1996) combine multiple local perceptrons in either a cooperative or a discriminative manner, and Zhang et al. (2006) combine multiple local support vector machines. These and other similar contributions rely on local neighborhoods containing multiple samples. The one-shot similarity kernel of Wolf et al. (2009) contrasts a single test sample with many training samples. We are unaware of any previous contribution that finetunes a model based on a single sample or any local learning approach that involves hypernetworks.

**Hypernetworks (Ha et al., 2016)** are neural models that generate the weights of a second *primary* network, which performs the actual prediction task. Since the inferred weights are multiplied by the activations of the primary network, hypernetworks are a form of multiplicative interactions (Jayakumar et al., 2020), and extend layer-specific dynamic networks, which have been used to adapt neural models to the properties of the input sample (Klein et al., 2015; Riegler et al., 2015).

Hypernetworks benefit from the knowledge-sharing ability of the weight-generating network and are therefore suited for meta-learning tasks, including few-shot learning (Bertinetto et al., 2016), continual learning (von Oswald et al., 2020), and model personalization Shamsian et al. (2021). When there is a need to repeatedly train similar networks, predicting the weights can be more efficient than backpropagation. Hypernetworks have, therefore, been used for neural architecture search (Brock et al., 2018; Zhang et al., 2019), and hyperparameter selection (Lorraine & Duvenaud, 2018).

MEND by Mitchell et al. (2021) explores the problem of model editing for large language models, in which the model's parameters are updated after training to incorporate new data. In our work, the goal is to predict the label of the new sample and not to update the model. Unlike MEND, our method does not employ the label of the new sample.

**Diffusion models** Many of the recent generative models for images (Ho et al., 2022; Chen et al., 2020; Dhariwal & Nichol, 2021a) and speech (Kong et al., 2021; Chen et al., 2020) are based on a degenerate form of the Focker-Planck equation. Sohl-Dickstein et al. (2015) showed that complicated distributions could be learnt using a simple diffusion process. The Denoising Diffusion Probablistic Models (DDPM) of Ho et al. (2020) extend the framework and present high quality image synthesis. Song et al. (2020) sped up the inference time by an order of magnitude using implicit sampling with their DDIM method. Watson et al. (2021) propose a dynamic programming algorithm to find an efficient denoising schedule and San-Roman et al. (2021) apply a learned scaling adjustments to the noise scheduling. Luhman & Luhman (2021) combined knowledge distillation with DDPMs.

The iterative nature of the denoising generation scheme creates an opportunity to steer the process, by considering the gradients of additional loss terms. The Iterative Latent Variable Refinement (ILVR) method Choi et al. (2021) does so for images by directing the generated image toward a low-resolution template. A similar technique was subsequently employed for voice modification Levkovitch et al. (2022). Direct conditioning is also possible: Saharia et al. (2022) generate photo-realistic text-to-image scenes by conditioning a diffusion model on text embedding; Amit et al. (2021) repeatedly condition on the input image to obtain image segmentation. In voice generation, the mel-spectrogram can be used as additional input to the denoising network Chen et al. (2020); Kong et al. (2021); Liu et al. (2021a), as can the input text for a text-to-speech diffusion model Popov et al. (2021).

## 3 PROBLEM SETTING AND ANALYSIS

We are given a dataset $S$ of samples $x_i \in \mathcal{X}$, for $i = 1..n$, and the associated labels $y_i \in \mathcal{Y}$ sampled i.i.d from some distribution $\mathcal{P}_{X \times Y}$ over the composite domain $\mathcal{X} \times \mathcal{Y}$. We consider prediction models $f : \mathcal{X} \times \Theta \to \mathcal{Y}$ that are parameterized by weight vectors in the domain $\Theta$. Specifically, we first learn a base model $f_\theta(x) = f(x, \theta)$, which is trained over the entire training set $S$.

For every sample $s \sim \mathcal{P}_{X \times Y}$, we further consider a finetuned version of $f_\theta$ that is optimized to overfit on the sample $s = (x, y)$, i.e., we minimize the loss of the single sample $s$, initializing the optimization process with $\theta$. We denote the obtained parameters as $\theta_s \in \Theta$ and the obtained network as $f_{\theta_s}$. Naturally, the prediction error of $f_{\theta_s}$ on the sample $s$ is expected to be small, improving upon that $f_\theta$ for that specific sample.

The meta-learning problem we consider is the one of learning a model $g$, that maps $x$ (the input domain of sample $s$) and potentially multiple latent representations of $x$ in the context of $f_\theta$, collectively denoted as $I(x)$, to a vector of weight differences, such that the following loss is minimized

$$\mathcal{L}(s) = MSE(\theta_s, \theta + g(x, I(x))), \tag{1}$$

where $MSE$ is the mean squared error. If $g$ generalizes well to **unseen** samples $s^* = (x^*, y^*)$, we would expect $f(x^*, \theta + g(x^*, I(x^*)))$ to be a better prediction of $y^*$ than $f(x^*, \theta)$.

We rewrite the mapping function $g(x, I(x))$ as a unit-norm component $g_u(x, I(x))$, and a scale factor $\rho(x, I(x))$, i.e.,

$$g(x, I(x)) = \rho(x, I(x)) \cdot g_u(x, I(x)), \text{ where} \tag{2}$$

$$||g_u(x, I(x))||_2 = 1 \tag{3}$$

Denote the mapping between a sample $s = (x, y)$ and $\frac{\theta_s - \theta}{||\theta_s - \theta||}$ as $H : \mathcal{X} \times \mathcal{Y} \to \Theta$. Let $H_1 : \mathcal{X} \times \mathcal{Y} \to \mathcal{X} \times \mathcal{Y} \times \Theta$ be the linear approximation of the operator $H$, i.e., $H(s + \delta s) = H(s) + H_1(s) \otimes \delta s + O(\delta s \otimes \delta s)$, where $\otimes$ denotes the tensor product along the sample dimensions.

Next, we provide two theoretical arguments without concrete quantization. First, following Kleinberg et al. (2018), if $\theta$ is near a local minima or an inflection point of the finetuning loss, then $\theta_s$ would converge to this point and $H(s + \delta s)$ is expected to be convex.

Second, since $\theta$ is obtained via an SGD optimization process over a sample from the distribution $\mathcal{P}_{X \times Y}$, and since $s \sim P_{X \times Y}$, it is near either a local minima or an inflection point for the training loss of finetuning with sample $s$. See, for example, Kleinberg et al. (2018).

Combining the two claims, we can expect $H$ to be locally convex.

**Lemma 1.** *Since $H$ is locally convex, it follows that the distribution $\mathcal{P}_{H_1(s)}$ in the domain $\Theta$ obtained when applying $H_1$ to samples $s \sim \mathcal{P}_{X \times Y}$ takes the following form:*

$$\mathcal{P}_{H_1(s)} = \mathcal{P}_{X \times Y}(H_1^{-1}(s) \otimes s)|det(H_1^{-1})|, \tag{4}$$

*where $H_1^{-1}(s) : (\mathcal{X} \times \mathcal{Y} \times \Theta) \times (\mathcal{X} \times \mathcal{Y})$ is the pseudoinverse of the tensor $H_1(s)$ (first we compute $H_1$ at given $s$, and then compute the tensor pseudoinverse).*

See appendix A for the proof.

From the lemma, the distribution of $H_1$ is a linear transformation of the distribution of $s$. Ho et al. (2020) showed that a diffusion process is suitable for estimating samples from complicated sample distributions. Assume that there exists a diffusion process, with weights $\gamma$, over the sample distribution, such that the variational bound is maximized

$$\max_\gamma E_q(log(p_\gamma(x_{0:T})) - log(q(x_{1:T}|x_0))), \tag{5}$$

where $p_\gamma$ is the model distribution, $q(x_0)$ is the data distribution, and $x_{0:T}$ forms a Markov chain. From Lemma. 1 we have that the image of the distribution of $H_1$ is contained in the image of $q$, with the same input set. Thus, if a diffusion model has the capacity to capture $q$, it will also capture $H_1$, which is a subset of $q$.

## 4 METHOD

Our method is based on a modified diffusion process. Recall that we denote the training dataset as $S = \{(x_i, y_i)\}_{i=1}^n$, and that for every training sample $s \in S$ we run fine-tuning based on that single sample to obtain the overfitted parameters (function) as $\theta_s$ ($f_{\theta_s}$). In our method, $\theta_s = \theta + g(x, I(x))$, where $\theta$ are the base model's parameters which are trained over $S$, and $g(x, I(x))$ is a mapping function that maps the input, i.e., the $x$ part of $s$, and multiple latent representations of it, $I(x)$, to the desired shift in the model parameters.

**Layer selection**    Current deep neural networks can have millions or even billions of parameters. Thus, learning to modify all network parameters can be a prohibitive task. Therefore, we opt to modify, via function $g$, on a single layer of $f_\theta$.

To select this layer, we follow Lutati & Wolf (2021) and choose the layer that presents the maximal entropy of the loss, when fixing the samples $s$, and perturbing the layer's parameters. Denote the perturbed weights, in which only layer $L$ is perturbed, as $\theta^L$. The score used for selection is

$$\sum_{(x,y)\in S} \mathrm{Entropy}_{\theta_L}\left(\mathcal{L}(f(x, \theta^L), y)\right),\tag{6}$$

where $\mathcal{L}$ is the loss objective on which the function of $f_\theta$ is trained on, and the entropy is computed over multiple draws of $\theta^L$. The entropy is computed by fitting a Gaussian Kernel Density Estimation (GKDE) (Silverman, 1986) to the obtained empirical distribution of the loss function. Since sampling does not involve a backpropagation computation, the process is not costly, so $10,000$ samples are used.

**The conditioning signal**    The latent representations, $I(x)$, has three components. Given a selected layer, $L$, we denote the input to this layer (when passing a sample $x$ to $f(x, \theta)$), as $i_L(x)$ and the activation of this layer as $a_L(x)$. We also use the output of the base function $f_\theta(x)$. $I(x)$ is, therefore, the tuple

$$I(x) = [i_L(x), a_L(x), f_\theta(x)]\tag{7}$$

### 4.1 DIFFUSION PROCESS

The diffusion goal is to reconstruct the mapping function $g_u(x, I(x))$. The process iteratively starts a random $\Omega_T$, iterates with $\Omega_t$, where $t$ is decreasing and is the diffusion step, and returns $\Omega_0$.

The diffusion error estimation network, $\epsilon_\Omega$ is a function of the current estimation, $\Omega_t$, the latent representation tuple, $I(x)$, and the diffusion timestep, $t$. The latter is encoded through a positional encoding network (Vaswani et al., 2017), $PE$. All inputs, except for $\Omega_t$ are combined into one vector: $e = PE(t) + E_i(i_L) + E_a(a_L) + E_o(f_\theta(x))$, where $E_i, E_a, E_o$ are the encodings of the layer input, layer activations and network output. Note that most of the components of $e$ do not change during the diffusion process, and can be computed only once. This way, the conditioning overhead is reduced to minimum. The conditional diffusion process is depicted in Fig 1.

**Training Phase**    The complete training procedure of $\epsilon_\Omega$ is depicted in Alg. 1. The first phase is overfitting, using simple gradient decent over a single input-output pair, see line 4. The overfitting phase is not demanding, since the backpropagation is conducted only over the selected layer and a single sample.

As stated in Sec. 4.2, while regular diffusion assumes that the input has unit variance, when estimating network weights, scaling has a crucial impact. This normalization ensures that the diffusion is trained over unit-variance input. We denote by $\theta_s^{\mathrm{norm}}$ the normalized difference between $\theta_s$ and the parameters $\theta$ of the base model (line 6).

Following Song et al. (2020), linear scheduling is used for the diffusion process, and $\beta_t, \alpha_t, \bar{\alpha}_t, \tilde{\beta}_t$ are set in line 8. A training example is then sampled:

$$\Omega_t = \sqrt{\bar{\alpha}_t}\theta_s^{\mathrm{norm}} + \sqrt{1 - \bar{\alpha}_t{}^2}\epsilon,\tag{8}$$

where $\epsilon \sim \mathcal{N}(0, 1)$ is normal noise. Since our goal is to recover the noiseless $\theta_s^{\mathrm{norm}}$, the objective is

$$||\epsilon - \epsilon_\Omega(\Omega_t, (I(x), t))||\tag{9}$$

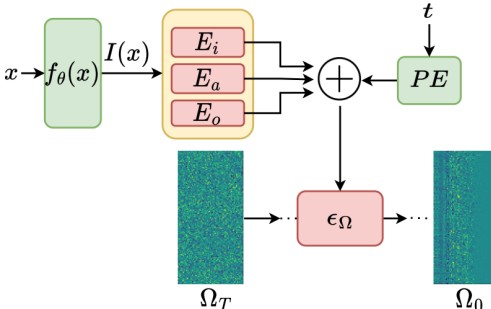

Figure 1: The diffusion process. $x$ is the input of the base network, $f_\theta(x)$. $I(x)$ is a tuple of latent representations of $x$. $E_i, E_a$, and $E_o$ are the input, activation, and output encoders, respectively, of the selected layer that is being modified. $t$ is the diffusion step, and $\Omega_t$ is the current diffusion estimation.

**Algorithm 1** Training Algorithm.
**Input:** $S$ training set, $\theta$ base network parameters, $\mathcal{L}$ the loss of the primary task, $T$ diffusion steps
**Output:** $\epsilon_\Omega$ diffusion network (incl. $E_i, E_a, E_o$).

1: **repeat**
2:     sample $(x, y) \sim S$ and set $\theta_s = \theta$
3:     **repeat**
4:         Grad step on $\nabla \mathcal{L}(y, f_{\theta_s}(x))$ to update $\theta_s$
5:     **until** $\mathcal{L}(y, f_{\theta_s}(x))$ converges
6:     $\theta_s^{\text{norm}} = \frac{\theta_s - \theta}{||\theta_s - \theta||}$
7:     $t \sim Uniform(1...T), \epsilon = N(\mathbf{0}, \mathbf{1})$
8:     $\beta_t = \frac{10^{-4}(T-t)+10^{-2}(t-1)}{T-1}, \alpha_t = 1 - \beta_t,$
        $\bar{\alpha}_t = \Pi_{k=0}^{k=t} \alpha_k$
9:     $\Omega_t = \sqrt{\bar{\alpha}_t} \theta_s^{\text{norm}} + \sqrt{1 - \bar{\alpha}_t}^2 \epsilon$
10:    Grad step on $\nabla ||\epsilon - \epsilon_\Omega(\Omega_t, (I(x), t))||$, updating $\epsilon_\Omega$, $PE$ and the components of $I$
11: **until** $||\epsilon - \epsilon_\Omega(\Omega_t, (I(x), t))||$ converges

A gradient step is taken in order to optimize this objective, see line. 10.

**Inference Phase** Given an unseen input $x$, $I(x)$ is computed using the base network $f(x, \theta)$ and is used for all calls to the diffusion network $\epsilon_\Omega$. The diffusion steps are listed in appendix B.

## 4.2 SCALE ESTIMATION

The Evidence Lower Bound (ELBO) used in Ho et al. (2020) assumes that the generated data has unit variance. In our case, in which the generated data reflects a difference in the layer's weights, the scale of the data presents considerable variation. Naturally, shifting the weights of a network by some vector $d$ or by some scale times $d$ can create a significant difference.

We, therefore, as indicated in Eq. 2, use an MLP network $\rho(x, I(x))$ to estimate the appropriate scale factor, based on the same conditioning signal that is used for the network $\epsilon_\Omega$ that implements $g_u$ as a diffusion process.

When learning network $\rho$, the following objective function is used

$$\mathcal{L}_{\text{scale}} = \sum_{s=(x,y) \in S} 10 \cdot log_{10}\left(\frac{|\rho(x, I(x)) - \rho_s|^2}{\rho_s}\right), \tag{10}$$

where $\rho_s = ||\theta_s - \theta||$.

## 4.3 ARCHITECTURE

The network $\epsilon_\Omega$ is a U-Net (Ronneberger et al., 2015), following Ho et al. (2020). Each resolution level has residual blocks, and an attention layer. The bottleneck contains two attention layers.

The positional encoder is composed of stacked sine and cosine encodings, following Vaswani et al. (2017). The encoders of $i_L, a_L$ are both single fully-connected layers, with dimensions to match the positional embedding. The encoder of the base network's output $f_\theta(x)$ depends on the output type. In the case of a classification network, where the output is a vector in $\mathbb{R}^C$, where $C$ is the number of classes, the encoder $E_O$ is a single fully-connected layer. In the case of image generation, the output image is first encoded using a VGG11 encoder (Simonyan & Zisserman, 2014), and then the latent representation is passed through a single fully-connected layer, again matching the dimension of the positional encoder. For speech separation, the estimated speech is first transformed to a spectogram with 1024 bins of FFT, then encoded using the same VGG11.

## 5 EXPERIMENTS

In all experiments the UNet $\epsilon_\Omega$ has 128 channels and five downsampling layers. The Adam optimizer (Kingma & Ba, 2014), with a learning rate of $10^{-4}$, is used. A linear noise schedule is used based on Song et al. (2020), and the number of diffusion steps is 10. All experiments are repeated three times to report the standard deviation (SD) of the success metrics. The inference overhead due to a U-Net forward pass is 27[ms] on a low-end Nvidia RTX2060 GPU (there are ten such passes, one per diffusion step).

In addition to the full method, we also show results for the network that overfits on the test data, which serves as an upper bound that cannot be applied without violating the train/test protocol. On some datasets we check to what extent selecting a single layer limits our results, by performing the overfitting process on all of the model weights. On all datasets, we ablate the scale component of our "Overfit with Conditional Diffusion models" (OCD) method, by estimating a fixed global scale factor $\bar{\rho} = \mathbb{E}_{s \in S}(\rho_s)$ as the mean value of the scale factor $\rho_s$ over the train-set. An additional ablation selects the model $f_{\theta_s}$ of the training sample $s$ with the closest $x$ to the test sample. This "nearest neighbor" ablation can be seen as the simplest way to implement the concept of OCD. Finally, we present an ablation that selects the layer with the second highest layer selection score, to evaluate the significance of the selection criterion.

**Image Classification**   Results for the MNIST dataset (LeCun & Cortes, 2010) are obtained with the LeNet5 architecture (Lecun et al., 1998). The selected layer is the one next to the last fully connected layer, which, as can be seen in appendix C has the maximal entropy among LeNet5's layers. CIFAR10 images (Krizhevsky et al., 2009) are classified using GoogleNet (Szegedy et al., 2015). The selected layer was the last fully-connected layer, see appendix C. For both architectures, the three encoders $E_{L_i}, E_{L_o}, E_O$ are simple fully-connected layers, with dimensions to match the number of channels in the UNet (128).

For classification experiments we measure both the cross entropy (evaluted on the test set) and the test accuracy. As can seen in Tab. 1, our method reduces the CE loss by a factor of 8 in comparison to the base network and there is a improvement of 0.5% in accuracy. Ablating the scale prediction, the results considerably degrade in comparison to the full method. The Nearest-Neighbor ablation yields slightly better results than the base network. The ablation that selects an alternative layer results in performance that is similar or slightly better than the base network. This is congruent with the small difference between fitting the selected layer and fitting all layers, which may suggest that much of the benefit of overfitting occurs in the selected layer.

On CIFAR10, our method improves classification accuracy from 92.9% to 93.7%. As in MNIST, much of the improvement is lost when running the three ablations. In both MNIST and CIFAR, when using the ground truth to overfit a specific example, the accuracy becomes, as expected, 100%. Considering the CE loss, overfitting the entire model instead of the selected layers yields only mild improvement (for MNIST below the standard deviation). This indicates that the added improvement gained by applying our method to all layers (and not just to the selected one) may not justify the additional resources required.

Experiments were also conducted on the TinyImageNet dataset (Le & Yang, 2015). The baseline network used is the (distilled) Data efficient transformer (DeiT-B/16-D) of Touvron et al. (2022). The selected layer is the weight of the head module. Our method is able to improve the baseline network, achieving 90.8% accuracy on test-set, falling short of the current state-of-the-art, which has double the parameters count, within 0.5%.

The weights in the OCD hypernetworks are obtained by a diffusion process, and multiple networks can be drawn for each sample $x$. When employing an ensemble of five classifiers using different initialization of the diffusion process noise, the results surpass the current state of the art, yielding 92.00% (4.7% better than the baseline model, and 0.65% better than the current state of the art).

**Image Synthesis**   We further tested our method on the image generation task of novel view synthesis, using a NeRF architecture (Mildenhall et al., 2021) and the "Synthetic-Blender" dataset. The Tiny-NeRF architecture (Murthy, 2020) employs an MLP network consisting of three fully-connected layers. The input is a 3D ray as a 5D coordinate (spatial location and viewing direction). The output is the corresponding emitted radiance. For each view, a batch of 4096 rays is computed, from which the interpolated image is synthesized.

Table 1: Performance on classification tasks. CE=Cross Entropy

| Method | MNIST (LeNet5) | | CIFAR10 (GoogleNet) | |
|---|---|---|---|---|
| | Test-CE ($\downarrow$) | Accuracy %($\uparrow$) | Test-CE ($\downarrow$) | Accuracy %($\uparrow$) |
| Base network | $0.080 \pm 0.009$ | $99.2 \pm 0.1$ | $0.085 \pm 0.01$ | $92.85 \pm 0.40$ |
| Overfitting on test | $0.002 \pm 0.0001$ | $100$ | $0.075 \pm 0.005$ | $100$ |
| Overfitting on test (All Layers) | $0.002 \pm 0.0001$ | $100$ | $0.073 \pm 0.003$ | $100$ |
| OCD nearest neighbor ablation | $0.073 \pm 0.010$ | $99.3 \pm 0.1$ | $0.082 \pm 0.02$ | $93.03 \pm 0.40$ |
| OCD no scaling ablation | $0.069 \pm 0.010$ | $99.3 \pm 0.1$ | $0.084 \pm 0.02$ | $93.01 \pm 0.35$ |
| OCD alternative layer ablation | $0.078 \pm 0.010$ | $99.2 \pm 0.1$ | $0.084 \pm 0.01$ | $92.96 \pm 0.27$ |
| OCD (ours) | $\mathbf{0.010 \pm 0.006}$ | $\mathbf{99.7 \pm 0.1}$ | $\mathbf{0.080 \pm 0.01}$ | $\mathbf{93.68 \pm 0.38}$ |

Table 2: Classification accuracy for TinyImageNet dataset.

| Method | Test-CE ($\downarrow$) | Accuracy %($\uparrow$) |
|---|---|---|
| Swin L/4 (Liu et al., 2021b) | NA | 91.35 |
| DeiT-B/16-D (Touvron et al., 2022) | 0.71 | 87.29 |
| DeiT-B/16-D + OCD | 0.65 | 90.80 |
| DeiT-B/16-D + OCD, ensemble of five | 0.65 | **92.00** |

Table 3: Performance (MSE±SD, lower is better) for the TinyNeRF network.

| Method | Lego | Hotdog | Drums |
|---|---|---|---|
| Base model | $0.076 \pm 0.004$ | $0.063 \pm 0.007$ | $0.068 \pm 0.006$ |
| Overfitting on test | $0.043 \pm 0.005$ | $0.032 \pm 0.005$ | $0.049 \pm 0.003$ |
| OCD no scaling ablation | $0.070 \pm 0.008$ | $0.060 \pm 0.005$ | $0.064 \pm 0.008$ |
| OCD (ours) | $\mathbf{0.052 \pm 0.006}$ | $\mathbf{0.047 \pm 0.004}$ | $\mathbf{0.057 \pm 0.006}$ |

We experimented with three objects from the dataset: Lego, Hotdog, and Drums. For each object a different TinyNeRF base model is trained over the corresponding training set. A single overfitting example is produced by considering a batch of 4096 rays from the same viewpoint.

Based on the data in appendix C, the first layer is selected. We, therefore, changed the layer-input encoder, $E_i$, such that the input image is first encoded by the VGG-11 encoder of Simonyan & Zisserman (2014) (pretrained over ImageNet-1k), followed by a fully-connected layer, to match the dimensions of UNet channels. The encoders $E_a, E_o$ are simple fully-connected layers, with dimensions to match the number of channels in the UNet (128).

As can seen in Tab. 3, our method improves the MSE by 31% on the Lego model, by 25% for Hotdog, and 16% for Drums. Without input-dependent scaling, the performance is much closer to the base network than to that of our complete method. Sample test views are shown in Fig. 2 and in Appendix D. Evidently, our method improves both image sharpness and color palette, bringing the synthesized image closer to the one obtained by overfitting the test image.

**Tabular Data** Gorishniy et al. (2021) have extensively benchmarked various architectures and tabular datasets. We use their simple MLP architecture as a base network (3 Layers). We were unable to reproduce the reported transformer, since the hyperparameters are not provided, and our resources did not allow us to run a neural architecture search, as Gorishniy et al. (2021) did. We run on two of the benchmarks listed: California Housing Kelley Pace & Barry (1997) (CA), which is the first listed and has the least number of samples, and Microsoft LETOR4.0(MI) (Qin & Liu, 2013), which is the last listed and has the largest number of samples.

Fig. 3(d) presents the layer selection criterion, with the first layer chosen for both datasets. As can seen in Tab. 4, for CA the base MLP model falls behind ResNet. Applying our method, the simple architecture achieves better results. For MI when applying our method, the simple baseline achieves a record MSE of $0.743$, surpassing the current best record on this dataset, which is 0.745 (Popov et al., 2020). The ablation that removes input-dependent scaling degrades the performance of the base network, emphasizing the importance of accurate scaling per sample.

Table 4: Tabular benchmarks by Gorishniy et al. (2021). $MSE \pm SD$, lower is better.

| Method | CA | MI |
|---|---|---|
| MLP | $0.4990 \pm 0.0030$ | $0.7470 \pm .0004$ |
| ResNet | $0.4860 \pm 0.0030$ | $0.7480 \pm .0003$ |
| Overfit MLP on test | $0.4750 \pm .0020$ | $0.7410 \pm .0003$ |
| OCD + MLP no scale | $0.5000 \pm .0030$ | $0.7490 \pm .0006$ |
| OCD + MLP (ours) | $\mathbf{0.4800 \pm .0020}$ | $\mathbf{0.7430 \pm .0004}$ |

Table 5: Performance of Gated-LSTM with the Hungarian loss on Libri5Mix.

| Method | SI-SDRi[dB] (↑) |
|---|---|
| Dovrat et al. (2021) | $12.7 \pm 0.1$ |
| Lutati et al. (2022) | $13.2 \pm 0.2$ |
| Overfit on test | $13.5 \pm 0.1$ |
| OCD no scale | $12.8 \pm 0.3$ |
| OCD (ours) | $\mathbf{13.4 \pm 0.1}$ |

Table 6: Classification accuracy for the NLP task of few-shot classification with 8 samples per class.

| Method | SST-5 | AmazonCF | CR | Emotion | EnronSpam | Average |
|---|---|---|---|---|---|---|
| T-few 3B (Liu et al., 2022) | **55.0** | 19.0 | 92.1 | **57.4** | **93.1** | 63.3 |
| SetFit (Tunstall et al., 2022) | 43.6 | 40.3 | 88.5 | 48.8 | 90.1 | 62.2 |
| SetFit + OCD | 47.8 | 41.0 | 90.5 | 50.2 | 92.2 | 64.3 |
| Ensemble of SetFit + OCD | 48.6 | **41.2** | **91.2** | 50.5 | 92.7 | **64.8** |

Table 7: The Cross-Entropy loss for the NLP task of few-shot classification with 8 samples per class using SetFIT + OCD, comparing one instance to the mean network obtained with 5 instances.

| Method | SST-5 | AmazonCF | CR | Emotion | EnronSpam |
|---|---|---|---|---|---|
| One instance | $0.45 \pm 0.19$ | $0.65 \pm 0.30$ | $0.41 \pm 0.11$ | $0.63 \pm 0.31$ | $0.39 \pm 0.20$ |
| Mean network of 5 instances | $0.42 \pm 0.20$ | $0.61 \pm 0.25$ | $0.39 \pm 0.11$ | $0.59 \pm 0.32$ | $0.38 \pm 0.19$ |

**Speech Separation**     To tackle the task of single microphone speech separation of multiple speakers, Nachmani et al. (2020) introduce the Gated-LSTM architecture with MulCat block and Dovrat et al. (2021) introduced a permutation-invariant loss based on the Hungarian matching algorithm, using the same architecture. Lutati et al. (2022) further improved results for this architecture, by employing an iterative method based on a theoretical upper bound, achieving state-of-the-art results.

The same backbone and Hungarian-method loss are used in our experiments, which run on the Libri5Mix dataset without augmentations, measuring the SI-SDRi score. The selected layer was the projection layer of the last MulCat block (appendix C). The output of the Gated-LSTM is the separated sounds, and to encode it, we apply the audio encoding described in Sec. 4.3 to each output channel separately and concatenate before applying the linear projection to $\mathbb{R}^{128}$.

As can seen in Tab. 5, applying our diffusion model over the Gated-LSTM model, we achieve $13.4dB$, surpassing current state-of-the-art results and approaching the results obtained by overfitting on the test data. The ablation that removes input-dependent scaling is much closer in performance to the base network than to our complete method.

**Prompt-free Few-Shot NLP Classification**     Recent few-shot methods have achieved impressive results in label-scarce settings. However, they are difficult to employ since they require language models with billions of parameters and hand-crafted prompts. Very recently, Tunstall et al. (2022) introduced a lean architecture, named SetFit, for finetuning a pretrained sentence transformer over a small dataset (8 examples per class), and then employing logistic regression over the corresponding embedding. In our experiments, we apply OCD to the last linear layer of the SetFIT sentence transformer. The U-Net has 64 channels, with 5 downsample layers. The size of the last linear layer is $768x768$. $I(x) \in \mathbb{R}^{768x1}$ is the embedding of the Sentence Transformer.

As can be seen in Tab. 6 using OCD and this lean architecture (with 110M parameters) outperforms, on average, the state-of-the-art model of Liu et al. (2022), which has 3B parameters. The mean performance across the datasets is improved by 1.0% over the state-of-the-art and by 2.1% over the SetFIT model we build upon. Recall that since the weights in the OCD hypernetworks are obtained by a diffusion process, one can draw multiple networks for each sample $x$. The variability that arises from the random initialization of the diffusion process allows us to use an ensemble. As can be seen in Tab. 6, when applying an ensemble of five inferences, the results of OCD further improve by 0.4% (on average) to a gap of 1.5% over the state-of-the-art.

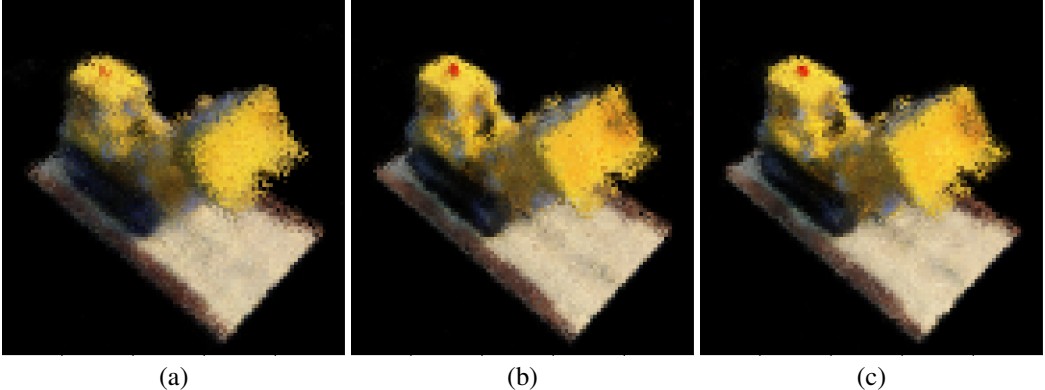

| (a) | (b) | (c) |

Figure 2: Sample TinyNeRF results (Lego model). More results can be found in Appendix D. (a) Base model on a test view. (b) Same test view, overfitted using the ground truth (c) OCD (ours).

Table 8: Runtime on a low-end Nvidia RTX2060GPU. Training includes the base model training, the collection of data for OCD by overfitting each sample in the training set, and the training of the diffusion model. The main overhead between the base model and the OCD one during inference is a result of running the UNet of the diffusion process for ten iterations.

| | Training[hrs] | | | Inference[ms] | |
|---|---|---|---|---|---|
| Architecture | Base | Overfit | Diffusion | Base | OCD |
| LeNet5 | 0.5 | 1.0 | 6.0 | 15 | 288 |
| GoogleNet | 2.5 | 1.0 | 5.0 | 20 | 298 |
| TinyNeRF | 1.1 | 0.2 | 5.2 | 230 | 510 |
| MLP | 1.2 | 0.1 | 6.2 | 75 | 355 |
| Gated-LSTM | 30 | 2.5 | 6.8 | 450 | 730 |
| SetFit | 0.2 | 0.1 | 5.1 | 125 | 401 |

Interestingly, when testing the average network weights of the 5 networks, the accuracy is virtually identical to the one obtained with a single OCD network (omitted from the table). However, there is some reduction in Cross-Entropy for the mean network (Tab. 7).

**Runtime**    Tab. 8 lists the measured runtime with and without OCD for both training and inference. Since overfitting each sample includes only 3 gradient descent steps over a single layer, most of the data collection results are shorter than the training time of the base model, with the exception of LeNet5 training that used a batch size of 16 versus batch size of 1 in the overfitting stage. Training the diffusion model is slower in most experiments than training the base model, except for the speech model, in which training the base model is lengthy. The inference overhead is almost constant across the six models since it consists of running the UNet $\epsilon_\Omega$ ten times. In absolute terms, the overhead incurred by OCD, even on a very modest GPU, is only a quarter of a second, while improving the results substantially. We note that the same $\epsilon_\Omega$ is used for all experiments, padding the actual weight matrix, which induces an unnecessary computational cost in the case of small networks.

## 6    CONCLUSIONS

We present what is, as far as we can ascertain, the first diffusion-based hypernetwork and show that learning the scale independently is required for best performance. The hypernetwork is studied in the specific context of local learning, in which a dynamic model is conditioned on a single input sample. This, too, seems to be a novel contribution. Using the diffusion architecture for the local learning task is justified by a first-order analysis of the change to a network's weights when performing limited finetuning.

The training samples for the hypernetwork are collected by finetuning a model on each specific sample from the training set used by the base model. By construction, this is only slightly more demanding than fitting the base model on the training set. More limiting is the size of the output of the hypernetwork, which for modern networks can be larger than the output dimensions of other diffusion processes. We, therefore, focus on a single layer, which is selected as the one that is most affected by weight perturbations. We extensively tested our method, tackling a very diverse set of tasks, using the same set of hyperparameters. We are yet to find a single dataset or architecture on which our OCD method does not significantly improve the results of the baseline architecture.

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

---

**Algorithm 2** Inference Algorithm.

**Input:** $x$ input sample, $\theta$ the parameters of the base network, $\epsilon_\Omega$ diffusion network, T diffusion steps.

**Output:** $g_u(x, I(x))$ estimated normalized $(\theta_s - \theta)$ for $s$ associates with $x$.

---

1: $t = T$
2: $\epsilon = N(\mathbf{0}, \mathbf{1})$
3: **while** $t \geq 0$ **do**
4: $\quad \beta_t = \frac{10^{-4}(T-t)+10^{-2}(t-1)}{T-1}, \alpha_t = 1 - \beta_t, \bar{\alpha}_t = \Pi_{k=0}^{k=t}\alpha_k, \tilde{\beta}_t = \frac{1-\alpha_{t-1}^-}{1-\bar{\alpha}_t}\beta_t$
5: $\quad \Omega_{t-1} = \frac{\Omega_t - \frac{1-\alpha_t}{\sqrt{1-\bar{\alpha}_t}}\epsilon_\Omega(\Omega_t, I(x), t)}{\sqrt{\alpha_t}} + \mathbf{1}_{t>1}\sqrt{\tilde{\beta}_t}$
6: $\quad t = t - 1$
7: **end while**
8: **return** $\Omega_0$

---

# A    PROOF OF LEMMA 1

**Lemma 1.** *Since $H$ is locally convex, it follows that the distribution $\mathcal{P}_{H_1(s)}$ in the domain $\Theta$ obtained when applying $H_1$ to samples $s \sim \mathcal{P}_{X \times Y}$ takes the following form:*

$$\mathcal{P}_{H_1(s)} = \mathcal{P}_{X \times Y}(H_1^{-1}(s) \otimes s)|det(H_1^{-1})|\,, \tag{11}$$

*where $H_1^{-1}(s) : (\mathcal{X} \times \mathcal{Y} \times \Theta) \times (\mathcal{X} \times \mathcal{Y})$ is the pseudoinverse of the tensor $H_1(s)$ (first we compute $H_1$ at given $s$, and then compute the tensor pseudoinverse).*

*Proof.* For any random variable $r$, and a mapping function $M$ that is monotonic and with a countable number of zeros, the following holds (Rosenblatt (1974)):

$$\mathcal{P}_{M(r)} = \mathcal{P}_r(M^{-1}(r))|\frac{dM^{-1}(r)}{dr}| \tag{12}$$

Where $M^{-1}$ is the pseudoinverse of $M$. From the local convexity of $H$, we have that its' Hessian is nonnegative.

$$\text{H is locally convex} \to \nabla^2 H(s + \delta s) \geq 0; \forall \delta s \to 0 \tag{13}$$

Recall that the Hessian of $H$ is the gradient of $H_1$, thus its gradient is non-negative, leading to monotonicity.

$$H_1(s) = \nabla H(s) \to \nabla H_1(s) = \nabla^2 H(s) \tag{14}$$
$$\nabla H_1(s) \geq 0 \tag{15}$$

In addition, since $H_1$ is a linear operator with finite size, it must have a finite dimension of null space.

$$Ker(H_1) \leq dim(H_1)\,, \tag{16}$$

where $Ker$ is the tensor null space, $dim$ is the tensor dimensions. This fulfills the second condition of the mapping function having a countable number of zeros. $\qquad\square$

# B    THE INFERENCE TIME ALGORITHM

The steps of the inference algorithm are listed om Alg. 2.

# C    LAYER SELECTION PLOTS

Fig. 3 depicts the layer selection criterion for various experiments.

# D    TINYNERF SAMPLE RESULTS

Fig. 4 presents sample results for the reconstruction of the Lego Model. Fig. 5 presents results for the Hotdog model. Fig. 6 presents sample test views of the Drums model.

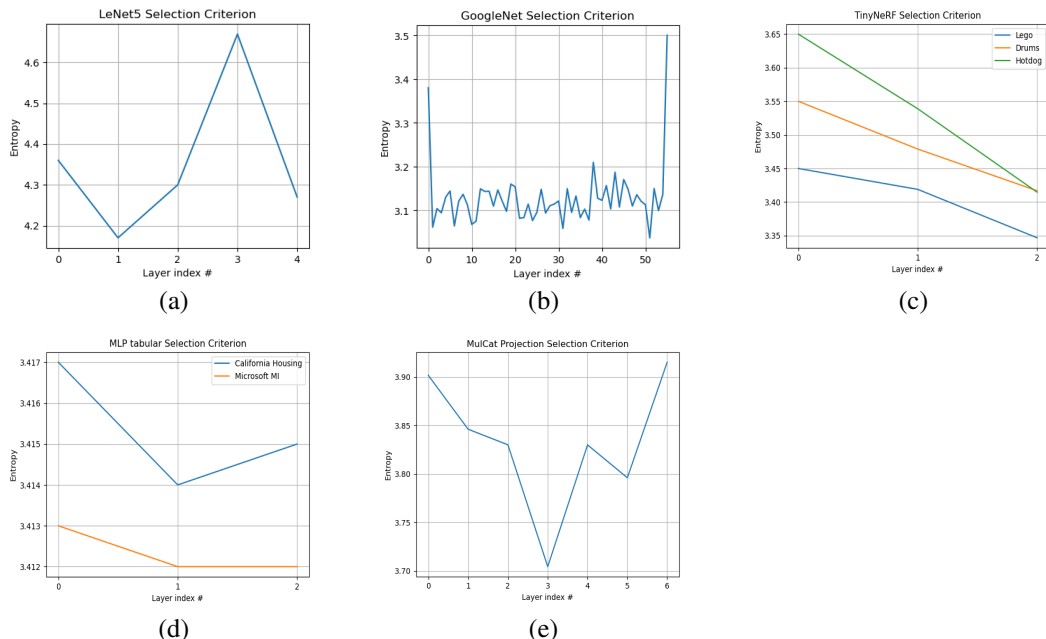

Figure 3: Layer Selection Criterion for different experiments. (a) For LeNet5 on MNIST, the next to last Fully-Connected layer is selected since it has the maximal entropy. (b) For GoogleNet on CIFAR10, the last Fully-Connected layer is selected. (c) For TinyNeRF (three datasets), the first Fully-Connected layer is selected. (d) For Tabular MLP the first layer is selected. (e) For MulCat the last projection layer is selected.

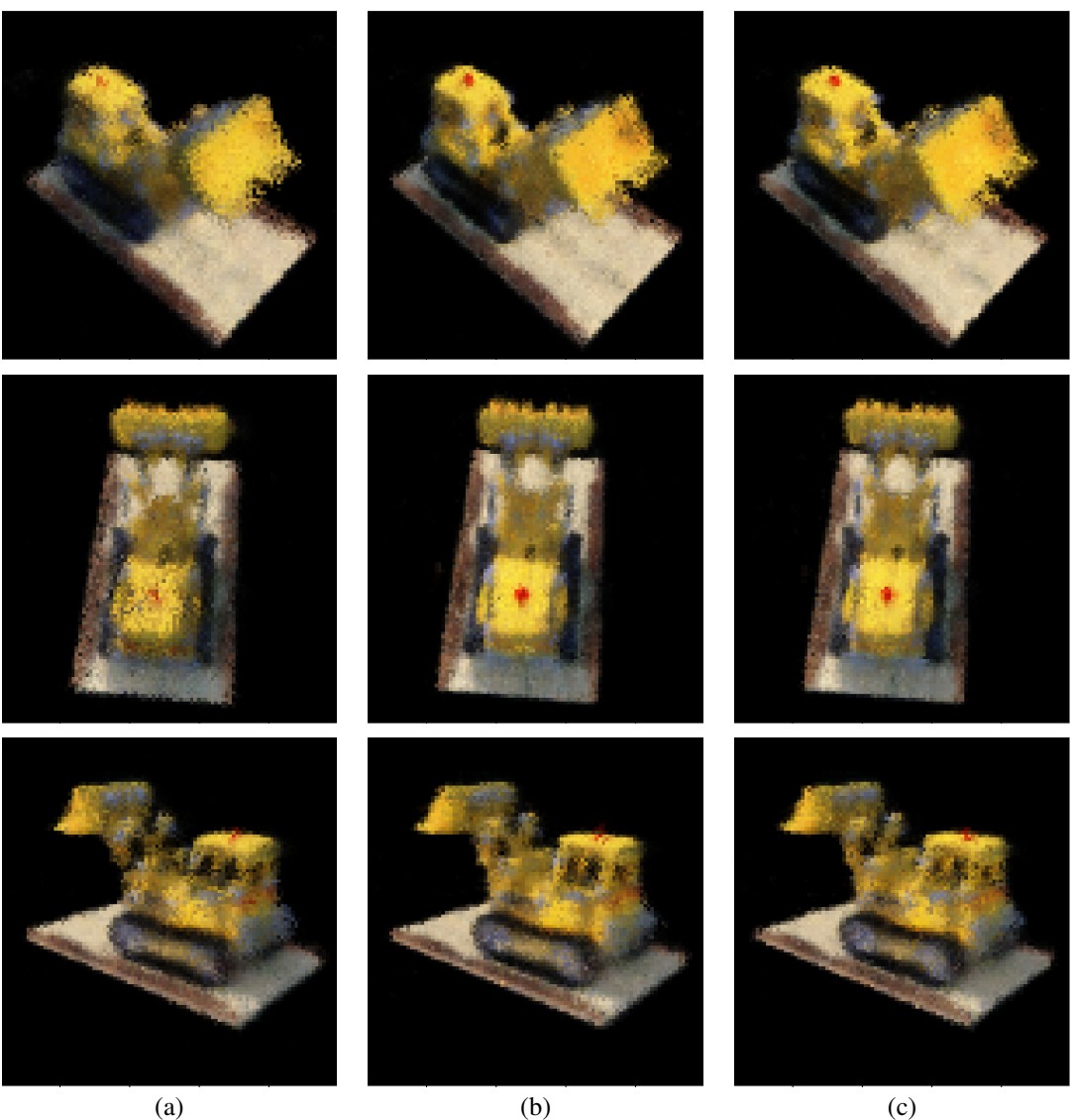

(a)     (b)     (c)

Figure 4: Sample TinyNeRF results on the Lego model. (a) Base model on a test view. (b) Same test view, overfitted using the ground truth (c) OCD (ours).

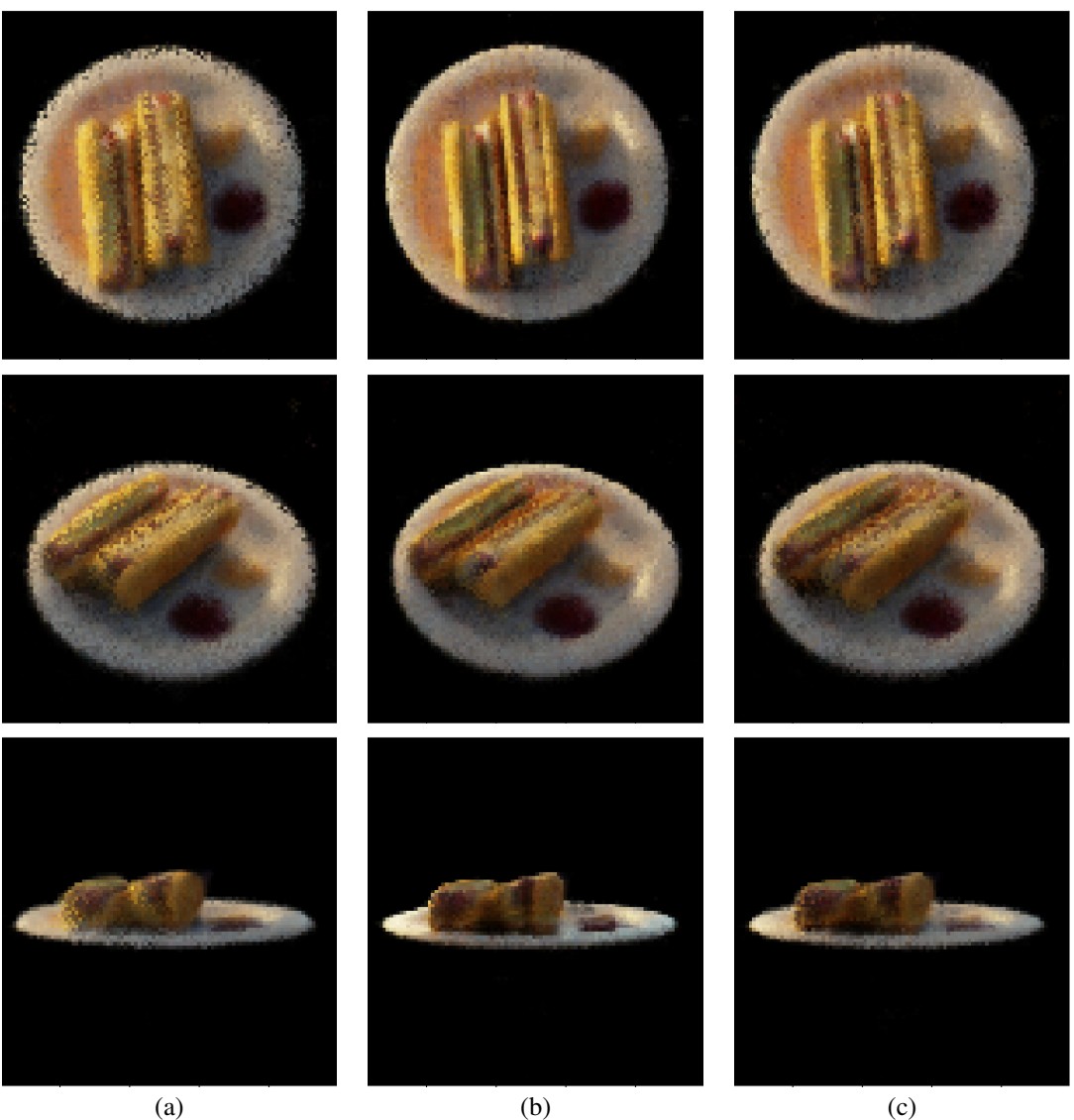

(a)           (b)           (c)

Figure 5: Sample TinyNeRF results on the Hotdog model. (a) Base model on a test view. (b) Same test view, overfitted using the ground truth (c) OCD (ours).

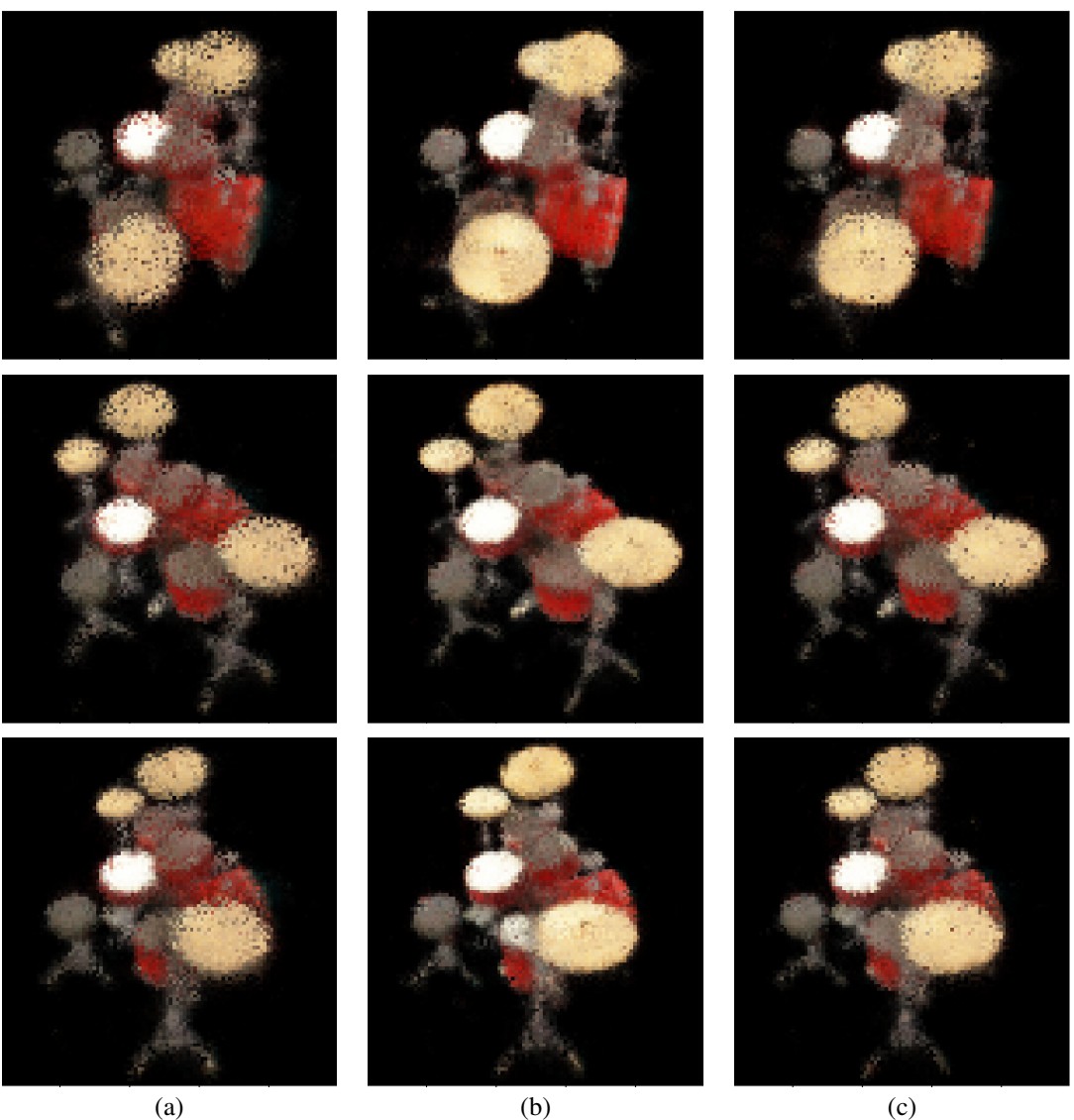

Figure 6: Sample TinyNeRF results on the Drums model. (a) Base model on a test view. (b) Same test view, overfitted using the ground truth (c) OCD (ours).

