# OpenReview forum: "OCD: Learning to Overfit with Conditional Diffusion Models"
_ICLR.cc/2023/Conference — Submitted to ICLR 2023_

### Official Review · Reviewer_Nu1A · 2022-10-23

**Confidence:** 3
**Correctness:** 4
**Technical Novelty And Significance:** 3
**Empirical Novelty And Significance:** 2
**Recommendation:** 5

**Clarity, Quality, Novelty And Reproducibility:**

Clarity and quality: A bit difficult to follow.

Novelty: Fair

Reproducibility: High

**Strength And Weaknesses:**

Stength
Using diffusion model for conditional parameter generation is novel.

Weakness
1. I find the paper a bit difficult to follow, it could use some rewriting/clearer plots to emphasize the motivation and approach.
2. I would like to see more details on the experiments: what's the cost of training and what's the overhead for inference.

**Summary Of The Paper:**

This paper proposes to use diffusion model to adapt selected layers of a trained model conditioned on an input sample. Authors conducted experiments onclassification, 3D construction, tablular data and speech separation.

**Summary Of The Review:**

This paper proposes to use diffusion model to generate/adapt certain weights of a trained model given a single input. The motivation and the training/inference pipeline could be better structured and explained.

---

> ### Author Response · Authors · 2022-11-05
> **Author response**
>
> We thank the reviewer for the feedback. It seems that the reviewer is not certain about the evaluation and we would be happy to work together toward improving the parts of our work that are unclear to the reviewer. We were challenged by the need to fit a comprehensive theory, a non-trivial method section, and a wide range of experiments, making the paper less verbose than we would have liked (please note, however, that reviewers A1Ss and Q7pB found the paper to be mostly clear). We would highly appreciate it if the reviewer could come back to us early with additional questions.
>
> Our motivation is given in the introduction, which positions our method as a modern approach to local learning that leads to an improvement across domains. The diffusion model approach is theoretically motivated in the analysis section. What is missing is to motivate local learning in general, which we added to the first paragraph of the related work in the revised version: “Local learning approaches perform inference with models that are focused on training samples in the vicinity of each test sample. This way the predictions are based on what is believed to be the most relevant data points. k-NN, for example, is a local learning method.”
>
> About the clarity of the approach and the plots, we would appreciate more specific feedback.
>
> Computational cost is not a limitation of our method, which is fairly efficient. Following the reviews, the revised version has additional details.
>
> Training -
> During training, we overfit each data point only once, by employing three finetuning steps using an SGD optimizer. The overfitted gradients are cached and used during training with random access. This data collection process depends on the size of the training dataset, for TinyNeRF it takes less than 1 minute, whereas for MNIST it takes 1 hour. The training of the U-Net takes an average of 7-8 GPU hours on a single, low-end RTX2060 GPU with 6GB of VRAM.
>
> Inference-
> As in the first paragraph in the experiments section indicates, for all experiments (including the new NLP ones), we use 10 diffusion steps. The time complexity of the U-Net network we employ with 128 channels and 5 downsampling blocks (same for all experiments since the weights are padded to fit 128x128 image) is 27 ms on an RTX2060 GPU.

---

> > ### Comment · Reviewer_Nu1A · 2022-11-17
> > **response**
> >
> > Thanks for the reply.
> >
> > I'm happy with the the new experiment details, even though it's now exceeding the suggested paper length.
> >
> > I had little doubt that "adapting" network weights for each sample would bring in extra performance gain, the question I had in mind was, why a diffusion model (could a more inference-time efficienct model suffice?), and is that a practical solution?
> >
> > For inference, 27ms overhead is per sample right ? For example for classifying 1k images, total overhead is 1k \times 27 ms?
> >
> > I'm rasining my score for the extra experiments and discussions.

---

> > > ### Author Response · Authors · 2022-11-17
> > > **Authors response**
> > >
> > > Thank you for raising the score. We would be happy to discuss any remaining concerns you may have.
> > >
> > > > paper length
> > >
> > > Naturally, we would fit the allocated length. During the discussion phase, we wanted to add without removing or compressing the existing material, so that changes are clearer. We would make sure to fit the page limit by, e.g., moving the proof and most panels of Figure 2 to the appendix.
> > >
> > > > why a diffusion model
> > >
> > > Using a hypernetwork that is based on the diffusion process:
> > > 1. It is justified by the theoretical analysis of Sec. 2, which shows that if the input domain can be generated by a diffusion model, then a diffusion model can be used for a hypernetwork implementing our local learning framework.
> > > 2. It is a key enabler (due to the randomized nature of diffusion processes) of the ensemble suggested by reviewer A1Ss, which, as we show, is extremely beneficial for real-world tasks.
> > > 3. While we cannot test all alternative hypernetwork architecture, further support for the utility of a diffusion process hypernetwork arises from our NLP experiments, since the SetFit network of Tunstall et al., 2022, which is used as the baseline network of the NLP experiments, is itself a hypernetwork.
> > >
> > > > 27ms overhead
> > >
> > > The 27ms mentioned are per sample and per iteration of the diffusion process. As can be seen in Tab. 8, since we use 10 diffusion iterations, the overhead per sample during inference is approximately 270ms.

---

> > > > ### Author Response · Authors · 2022-11-17
> > > > **page limit update**
> > > >
> > > > Thank you again for your feedback. Following your recent comment, we have rearranged the revised manuscript to fit within the page limit.

---

### Official Review · Reviewer_Q7pB · 2022-10-24

**Confidence:** 4
**Correctness:** 3
**Technical Novelty And Significance:** 4
**Empirical Novelty And Significance:** 3
**Recommendation:** 8

**Clarity, Quality, Novelty And Reproducibility:**

Clarity:
Mostly clear.

Quality:
See strengths and weaknesses.

Novelty:
AFAIK the author's claims are true that this is the first "local learning approach that involves hypernetworks" and the first "diffusion-based hypernetwork", and the authors make some well-motivated design choices.

Reproducibility:
The method is described clearly and code for OCD is attached, which together should be enough for reproducibility.

**Strength And Weaknesses:**

Strengths:
 - Strong experimental results on a wide variety of problems and neural architectures (image classification, NeRF, tabular data, speech separation). State-of-the-art on some of these.
 - Well-motivated and experimentally-justified innovations include predicting the scale of the weight change separately, and changing weights for only a single network layer.
 - Apart from some minor points listed below, the writing is clear and a pleasure to read.

Weaknesses:
- There is a lack of discussion of computational cost. Relative to the baselines they consider (mostly just the same networks as for OCD, but without weight adaptation), OCD is presumably considerably more costly at both training time and test time. Since at training time, it requires both performing many "finetuning" SGD steps, and training the diffusion model. And the test-time weight adaptations are probably expensive relative to the cost of a single forward pass. Ideally there would be a comparison where the baselines were given larger architectures, and trained for longer, etc., to match OCD for computational cost. I appreciate that this paper can still be a valuable contribution without "winning" in such a comparison but this would provide very helpful context. Failing this, the differences in training time and test time should at least be made explicit and discussed. I am willing to raise my score if you address this point.

Minor:
- Section 3, "$s$ [...] is near either a local minima or an inflection point for the training loss of finetuning with sample $s$". I don't think this is necessarily true. E.g. in image classification the loss would be minimised when the "correct" output logit is driven to infinity and so it seems that the minimum may be at infinity (or more precisely, there may not be any minimum) when training on a single data point.
- Equation 10. It is not clear what $x$ refers to here, or what the "data distribution" $q(x_0)$  refers to. The variable $x$ previously referred to inputs to the base model (i.e. in Eq. 1) but I think this might now be referring to a variable in the space of the network weights.
- Section 3, "without concrete quantization" - probably not important, but I cannot tell what is meant by this phrase

**Summary Of The Paper:**

The authors propose a diffusion model which, given a neural network input x, adapts the neural network weights to specialise the network to this single input. Doing so before evaluating on each test example (x, y) leads to better predictions of y. The diffusion model is trained to imitate an SGD procedure which overfits a pretrained model's weights to a single data point.

**Summary Of The Review:**

The method is novel and clearly-presented. The results are good, but their significance is a little unclear to me because of the lack of presentation and discussion of computational cost. I will consider raising my score if this is improved.

~~EDIT: Lowering my score; see "Additional related work" comment below.~~

---

> ### Author Response · Authors · 2022-11-05
> **author response**
>
> We thank the reviewer for the helpful remarks, and we hope our reply will fully address the raised concerns. If not, please let us know early so that we can provide more data.
>
> Computational cost is not a limitation of our method, which is fairly efficient. Following the reviews, the revised version has additional details.
>
> Training -
> During training, we overfit each data point only once, by performing three finetuning steps using an SGD optimizer. The overfitted gradients are cached and used during training with random access. This data collection process depends on the size of the training dataset, for TinyNeRF it takes less than 1 minute, whereas for MNIST it takes 1 hour. The training of the U-Net takes an average of 7-8 GPU hours on a single low-end RTX2060 GPU with 6GB of VRAM.
>
> Inference-
> As in the first paragraph in the experiments section indicates, for all experiments (including the new NLP ones), we use 10 diffusion steps. The runtime of the U-Net we employ, which has 128 channels and 5 downsampling blocks (same for all experiments since the weights are padded to fit a 128x128 image) is 27 ms on this RTX2060 GPU.
>
> Addressing the minor concerns raised by the reviewer :
>
> While the logit goes to infinity the loss goes to zero (whether it is the MSE loss between true label or BCE), our functional plane is the loss and not the logit, therefore it is indeed near local minima.
>
> The text below equation 10 is the explanation for x_{0:t}, formulating the Markov chain of the diffusion process. This notation is the same as in other diffusion papers.
>
> "Without concrete quantization" means that we do not write down the actual bounds as exact inequalities, since this would be overly technical, would require considerably more background, and would incur additional notation.

---

> > ### Comment · Reviewer_Q7pB · 2022-11-09
> > **response**
> >
> > Thanks for the extra information!
> >
> > Regarding the inference-time cost, could you put this into the context by stating how long the base model takes to run? I.e. if the 10 diffusion steps take 27ms each, and the base model takes $x$ms, then the total inference time of your method is presumably $(270+x)$ms. So the time relative to the base model is very different if $x$ is several thousand, as opposed to if $x$ is much less than 270. Having these overall runtimes clearly stated somewhere would be very helpful the reader decide whether the improvement from OCD is "worth" the additional cost.
> >
> > Similarly for the training-time costs, I would encourage you to add them to the paper somewhere, and contextualise them with the training time of the base network for each experiment.
> >
> > Finally, regarding the notation in Equation 10, I appreciate that using $x_{0:T}$ is common notation in diffusion papers. It is a little confusing here though as it clashes with the previous use of $x$ as the input to some model. I would suggest perhaps replacing $x_{0:T}$ here with e.g. $\theta_{0:T}$.

---

> > > ### Author Response · Authors · 2022-11-09
> > > **Thank you for your response!**
> > >
> > > We thank the reviewer for the quick reply, which is very helpful for us. We would be happy to address any additional concerns.
> > >
> > > Following the reviewer’s request, we added Table 5, which lists the runtime of (1) training the base model, (2) collecting the data for OCD, and (3) training the diffusion model, as well as that of the inference step with and without OCD. For the smaller models, the overhead may seem large in percentages, but it is modest in absolute numbers and we note that the UNet we used is the same across all experiments, which is a bit of an overkill for small networks.
> > >
> > > Regarding the notation around equation 10:
> > >
> > > In the sentence preceding the equation: "Assume that there exists a diffusion process, with weights $\gamma$, over the sample distribution, such that the variational bound is maximized" we make an assumption regarding the input of the base model, which is $x$. To be clear, this is not the diffusion process on the network parameters $\theta$.
> > >
> > > We then use Lemma 1 to show that the image of the diffusion model over $x$ contains the image of the gradients over $x$ as it is approximated by the linear transformation. Thus we conclude that there exists a diffusion model that can reconstruct the gradients needed to finetune the model conditioned on the input $x$.
> > >
> > > We would be grateful if the reviewer could take our answers into consideration. We believe that the paper now addresses all the raised concerns and that thanks to the reviewers it has improved since the initial submission. It is now much clearer, has more details, and obtains state-of-the-art results in an NLP task, for which there are very recent contributions by leading NLP teams.

---

> > > > ### Comment · Reviewer_Q7pB · 2022-11-16
> > > > **Raising my score**
> > > >
> > > > Thank you for your response, and for adding the table. I am satisfied that my concerns have been addressed, and as far as I can tell the issues raised by the other reviewers have also been addressed. I have raised my score to an 8.

---

> ### Comment · Reviewer_Q7pB · 2022-12-11
> **Additional related work**
>
> During the reviewer-AC discussion, Reviewer wwSB mentioned some additional related work: [Tent: Fully Test-time Adaptation by Entropy Minimization](https://arxiv.org/abs/2006.10726) and [Tailoring: encoding inductive biases by optimizing unsupervised objectives at prediction time](https://arxiv.org/abs/2009.10623). Both of these seem very relevant (as they involve finetuning given a specific input without seeing the corresponding ground-truth output) and I think should be compared against. I am therefore lowering my score to a 5. Apologies that this has come up after the end of the discussion period; I was previously unaware of these papers and largely agreed with the author's responses regarding the related work that was previously mentioned on this page.

---

> > ### Author Response · Authors · 2022-12-11
> > **Authors' response**
> >
> > Thank you for your note and for acknowledging the challenging timing of it. Given that it involves at least two reviewers, we have replied to it as an official comment to all reviewers.

---

> > > ### Author Response · Authors · 2022-12-12
> > > **important update**
> > >
> > > We have now added the requested empirical results, which were the reason for the last-minute change in the paper's score. Thank you for your hard work.

---

> > > > ### Comment · Reviewer_Q7pB · 2022-12-12
> > > > **Thank you**
> > > >
> > > > Thank you for the additional results. I think they make a convincing case that the prior work is not designed to tackle the same problem and, in any case, does not work as well as OCD when applied to the same problem. I hope the authors will include these empirical results in the paper, and discuss the differences from these papers in the related work. I am raising my score back to an 8.

---

> > > > > ### Author Response · Authors · 2022-12-13
> > > > > **Thank you for helping us improve our work**
> > > > >
> > > > > As requested, we will discuss the two local learning contributions you pointed to in your comment from Dec 11 and add the latest empirical results that were posted.

---

### Official Review · Reviewer_A1Ss · 2022-10-25

**Confidence:** 3
**Correctness:** 3
**Technical Novelty And Significance:** 3
**Empirical Novelty And Significance:** 3
**Recommendation:** 6

**Clarity, Quality, Novelty And Reproducibility:**

- The paper is well-written for the most part.
- The presented method is novel.
- Code is provided in Supplementary.

**Strength And Weaknesses:**

## strengths
- The paper presents an interesting idea of using conditional diffusion model as a hypernetwork generator.
- The experiments covers a wide range of applications including image classification, 3-D reconstruction, tabular data and speech separation.
## weakness and questions
- The paper is well-written for the most part but some parts can be improved to make the manuscript clearer. For example:
	- In Eq. 1, $\theta$ is not formally defined before introduction. Is $\theta$ the "base network parameters"? Does this mean the initialized weights or the pretrained weights?
	- What are $\Omega$'s in Sec 4.1? Are they of the same dimensionality as $\theta$?
	- The presentation of figures can be improved. For example, Figure 2 is a little hard to read, the lines are very thin and captions/ticks/legends are very tiny.
- UNet: What's the dimensionality of $\Omega$? If we choose a conv layer then should we use a 4-D UNet? In the case shown in Figure 1, is it true that the weight matrix is treated/reshaped as a 2-D "image" and we operate a 2-D UNet on it? If so, any explanation for why the inductive bias introduced by UNet is useful here?
- The experiments on image classification are a little bit limited. It would be better to test the algorithm on larger networks and larger datasets e.g. TinyImageNet.
- At the inference phase, do we get a single point estimate of $\theta$? Is it possible to employ an ensemble here? With the proposed OCD formulation, it might be interesting to integrate/compare the Bayesian ensemble here.

**Summary Of The Paper:**

The paper proposes a hypernetwork model where the weights are conditioned on input sample $x$ and are trained to match the model after finetuning. The hypernetwork generator is a conditional diffusion model, conditioned on $x$ (and it's features), and operates in the space of network weights $\theta$.

**Summary Of The Review:**

I think it is an interesting paper, although the presentation is not clear enough. I am willing to amend my scores if my concerns are addressed.

---

> ### Author Response · Authors · 2022-11-05
> **author response**
>
> We thank the reviewer for the valuable feedback. We hope this reply will address the concern in full. If not, please let us know early.
>
> θ is the vector of the parameters of the base network, which are pretrained. This is explained in the first paragraph of “Problem Setting and Analysis“: “Specifically, we first learn a base model fθ (x) = f (x, θ), which is trained over the entire training set S”.
>
> Ω is the diffusion process, as stated in the first paragraph of subsection 4.1:
> “The diffusion goal is to reconstruct the mapping function g_u(x, I(x)). The process iteratively starts a random Ω_T , iterates with Ω_t, where t is decreasing and is the diffusion step, and returns Ω_0.” The diffusion process has the same dimensionality as θ.
>
> U-Net: As stated in the first experiment paragraph, “UNet ε_Ω has 128 channels and five downsampling layers”.
>
> ConvLayer: Indeed we need to employ a 4D U-Net. It is true that the weight is treated as an image, as the reviewer states. We use U-Net in order to accommodate the connections there are between different entries in the weight matrix. When using a simple MLP the training failed to converge without hand-crafted dropout. We found empirically that the U-Net architecture was robust enough for all benchmarks of all tasks, with the exact same set of hyperparameters.
>
> Ensemble: This is a very interesting idea, and should be looked at as future work. Currently, there is one set of parameters that are issued by the hypernetwork, although it is easy to obtain multiple ones due to the diffusion process.
>
> Regarding larger datasets: we preferred to address multiple domains and in non-vision domains, we have fairly large datasets. In speech separation, our experiments exhibit state-of-the-art results on large datasets. The revised version also includes experiments in a new domain, namely that of NLP. Our solution provides state-of-the-art results using a transformer model with only 110M parameters and surpasses a recent Transformer with 3B parameters (we add OCD to a solution that was published in September that is almost state of the art).

---

> > ### Author Response · Authors · 2022-11-07
> > **Ensembles of OCD networks**
> >
> > We have just posted a 2nd revision in which we share the results of applying an ensemble to the NLP experiment. As it turns out, there is a significant improvement when using an ensemble of 5 OCD networks. The networks differ only by the random initialization of the diffusion process.
> >
> > Thank you for the very helpful suggestion!

---

> > > ### Author Response · Authors · 2022-11-15
> > > **TinyImageNet experiment**
> > >
> > > Following your request, we present results on TinyImageNet. Since we were not able to replicate the baseline results of the SOTA network (Swan L/4), we chose to work with DeiT-B/16-D. As shown in Tab. 2 of the revised version, we were able to improve the results of DeiT by a wide margin from 87.3% (our runs) to 90.8%. Even more exciting, using your suggestion to use an ensemble of OCD networks with random initialization of the diffusion process, we were able to obtain 92.0%, which is 4.7% better than the baseline DeiT-B/16-D network, surpassing the current SOTA of 91.4%.

---

> > > > ### Comment · Reviewer_A1Ss · 2022-11-19
> > > > **Response to the authors' rebuttal**
> > > >
> > > > Sorry for the late response. I thank the authors for their diligent efforts in the rebuttal. It was nice to see that the authors could conduct more experiments to improve their work and make relevant changes to their paper. I have increased my score based on the author's responses.

---

> ### Comment · Reviewer_A1Ss · 2022-12-07
> **Additional comments**
>
> Just realized there are some typos in the paper, e.g.:
> - $H_1^{-1}:\mathcal{X}\times\mathcal{Y}\times\Theta \mapsto \mathcal{X}\times\mathcal{Y}$
> - In both Algorithm 1 line 9 and Eq. (8), $\sqrt{1-\bar{\alpha}^2}$ should be $\sqrt{1-\bar{\alpha}}$
>
> Also, the connection between the theoretical argument and the methodology seems weak. It would be good if the authors could fix the typos and add more analysis/discussion on how Lemma 1 motivates the use of diffusion models here.

---

> > ### Author Response · Authors · 2022-12-08
> > **Authors response**
> >
> > Thank you for making an additional pass over our manuscript. Your suggestions have greatly contributed to our work and have demonstrated the value of the openreview model.
> >
> > ### Typos
> >
> > Thank you for pointing out these typos.
> > * We will fix the domains of $H_1^{-1}$.
> > * Indeed the square sign should be removed.
> >
> > ### The connection between the theory we present and the methodology
> >
> > Following the review, we would add text based on the discussion below.
> >
> > Our contribution is, as far as we can ascertain, the first hypernetwork that employs a diffusion model. We, therefore, take upon ourselves to show two crucial points: (i) that diffusion models are able to predict the weights of a network in our setting, and (ii) how to employ a diffusion model as the basis of a hypernetwork.
> >
> > For the first point, it is no doubt that diffusion models provide a strong tool for sampling from many conditional distributions. However, the question of which distributions can be sampled in this way is often neglected. Sometimes, there is a clear motivation for using diffusion models, e.g., in deblurring images, or, when applying them to error-correcting-codes (https://openreview.net/forum?id=rLwC0_MG-4w). However, often there is no such motivation and the usage of diffusion models is based on empirical success.
> > In our work, we make a novel type of argument and prove that if a diffusion model exists for the input domain of the primary learning problem (i.e., the problem that OCD is applied to) conditioned on the output domain, then OCD can employ a diffusion model.
> >
> > Lemma 1 suggests that the difference between the baseline weights to the finetuned version is a random variable with a distribution that is a linear transformation of the data distribution. Therefore, based on the assumption that the primary learning task's data can be modeled by a diffusion model, a diffusion model can predict the difference of the weights in the OCD framework.
> >
> > The theory also provides a crucial observation regarding the second question (how to use a diffusion model as an OCD hypernetwork). Namely, the diffusion model should be applied to the difference between the baseline network and the finetuned one. Following the review, we will add to the supplementary material convergence graphs showing that training the diffusion hypernetwork to predict the weights (and not the differences) is not feasible.

---

### Official Review · Reviewer_wwSB · 2022-10-25

**Confidence:** 4
**Correctness:** 2
**Technical Novelty And Significance:** 2
**Empirical Novelty And Significance:** 2
**Recommendation:** 3

**Clarity, Quality, Novelty And Reproducibility:**

The problem tackled by the paper seems interesting, but I think the exposition of the paper could be significantly improved to describe the exact benefits and methodology in the paper.

**Strength And Weaknesses:**

# Strengths

**Novelty.** The problem studied by the paper is interesting and is new to my knowledge

# Weaknesses

**Clarity.** The paper is somewhat poorly written -- the introduction does not motivate the problem to be studied, related works are incorrectly referenced and the method section is confusing and does not adequately describe what is happening in the approach. What is the train data used for training the method? How is it obtained? How is the diffusion process used to obtain the data necessary to overfit data?

**Results.** The results provided the paper are somewhat toy -- the NeRF look very poorly fit for example and the paper does not appear to be empirically rigorous. The paper does not explain why the approach only works on a single layer in the network. Also I could not completely understand the different baselines presented in each table. Why is each baseline natural?

**Significance.** The results provided in the paper are not significant in its current state. The proposed approach seems to substantially underperform existing baselines -- the paper could be improved substantially by illustrating an concrete application of the approach on a real world setting that outperforms existing baselines.

==============
Post Rebuttal Update

I have decided to downgrade my score (see the discussion below for some issues raised). The results for instance on TinyNeRF seem incorrect (even with 32 ray samples the results are still much sharper than those shown in the paper) and generally seem toy. This problem setup seems very similar to existing meta-learning techniques, for instance a meta-learned loss https://arxiv.org/abs/1906.05374 can be used to adapt the model (where we can not input a label y into the learned loss function). Experiments are run only with very limited data on very small architectures.

**Summary Of The Paper:**

This paper presents an approach to use diffusion models as a manner to parameterize a meta-learning algorithm, where the network perbutation prediction by the diffusion process adapts a base model to a separate task.

**Summary Of The Review:**

I think the paper is a bit preliminary -- a lot of work needs to go into the writing to clarify the paper and results could be significantly improved.

---

> ### Author Response · Authors · 2022-11-05
> **author response**
>
> We thank the reviewer for the detailed feedback. We believe that our reply fully addresses all raised concerns and would highly appreciate it if the reviewer could come back to us early with additional questions.
>
> Motivation in the introduction: we believe that the introduction clearly positions our method as a modern approach to local learning that leads to an improvement across domains. What is missing is to motivate local learning in general. We, therefore, add the following text to the first paragraph of the revised related work section:
> “Local learning approaches perform inference with models that are focused on training samples in the vicinity of each test sample. This way, the predictions are based on what is believed to be the most relevant data points. k-NN, for example, is a local learning method.”
>
> Incorrect references: we had many issues due to relying on a bib source that put arxiv references in a format that is not compatible with the bibliography format that was used. We also had several other mistakes. These are now fixed.
>
> Confusing method section: Clearly we were challenged by the need to fit a comprehensive theory, a non-trivial method section, and a wide range of experiments. Please also note that reviewers A1Ss and Q7pB found the paper to be mostly clear.
>
> To your specific questions about the method section:
> > What is the train data used for training the method? How is it obtained?
> We learn to map a sample to a vector of network weights that is obtained when overfitting on that sample. To obtain the training dataset for this mapping, we fine-tune the base model on every single sample in the training set, one sample at a time.
>
> This is defined in the problem setting section and we now make it clearer in the recap at the beginning of the method section by adding the sentence “...that for every training sample $s\in S$ we run fine-tuning based on that single sample to obtain…”
>
> > How is the diffusion process used to obtain the data necessary to overfit data?
> We are not sure what is meant by this question, sorry. Please clarify.
>
> As for the results: we provide empirical evidence in four different domains, and now, following the reviewer’s request also in NLP. In two very different cases (speech and NLP) we are the absolute state of the art, satisfying the reviewer’s call for “a concrete application of the approach on a real-world setting that outperforms [all] existing baselines.” Specifically, in speech (table 4), our method is state-of-the-art in the speech separation problem, which is an extensively researched domain.
>
> The NLP experiments we present build upon the very recent (Sep. 2022) preprint by Tunstall et al. “Efficient Few-Shot Learning Without Prompts”. This recent work introduces a transformer model called SetFIT, which has 110M parameters. This is much fewer than the 3B parameters of the T-Few 3B model  (Liu et al., August 2022), which is the current SOTA in this type of few-shot learning.
>
> Beyond demonstrating our work for NLP, these experiments also verify its performance on a transformer model. The selected layer is the linear layer of the last transformer layer of the SetFit model.
>
> As the table below indicates, when using OCD, the average score over all the datasets surpassed the SOTA by 1.02% and the SetFit method we are based on by almost 2.1%.
>
> |            | SST-5 | AmazonCF | CR   | Emotion | EnronSpam | Average |
> |------------|-------|----------|------|---------|-----------|---------|
> | T-Few 3B   | 55    | 19       | 92.4 | 57.4    | 93.1      | 63.32   |
> | SetFit     | 43.6  | 40.3     | 88.5 | 48.8    | 90.1      | 62.26   |
> | SetFit+OCD | 47.8  | 41       | 90.5 | 50.2    | 92.2      | 64.34   |
>
>
> > The paper does not explain why the approach only works on a single layer in the network.
>
> The method can very naturally work on multiple layers. However, it is more efficient to run on a single layer. Since modifying one layer is sufficient to obtain improved performance, it makes sense to employ just a single layer.
>
> This is explained in the introduction (“we [automatically] select a specific layer of the neural model and modify only this layer. This considerably reduces the size of the generated data and, in our experience, is sufficient for supporting the overfitting effect.“) and the entire 2nd paragraph of the method section.
>
> Additionally, we perform suitable ablations, as mentioned in the experiments section: “On some datasets [where it is computationally feasible] we check to what extent selecting a single layer limits our results, by performing the overfitting process on all of the model weights.” These experiments lead to the conclusion that “...overfitting the entire model instead of the selected layers yields only mild improvement (for MNIST below the standard deviation). This indicates that the added improvement gained by applying our method to all layers (and not just to the selected one) may not justify the additional resources required.”

---

> > ### Comment · Reviewer_wwSB · 2022-11-26
> > **Reviewer Response**
> >
> > I thank the authors for their detailed comments, but at the moment I do not think the paper is suitable for publication in its current form and maintain my score:
> >
> > Here are a couple issues that concern me:
> >
> > 1) The underlying performance of the method seems quite limited -- for instance the images presented in Figure 2 are very blurry and much
> > worse than even the original results in the tinynerf codebase (https://github.com/bmild/nerf/blob/master/tiny_nerf.ipynb). I don't understand why the images of the baseline method are so poor when presented in the paper. Could it be that compared baselines are not fully trained? That would be a significant issue with the paper.
> > 2) I would like to see some analysis on the effect of different layers of finetuning -- it seems like for a wide range of tasks, having more than one hidden layer finetuned would be important and this experiment does not seem to conflict with stated goal of local learning.
> > 3) Most of the networks considered in the paper are very small -- much smaller than what is commonly used (for example in image classification or NeRF overfitting). For these results to be more generally useful, it would be good to demonstrate efficacy on more modern architectures. Similarily, each of the dataset evaluated (for example the NLP setting) use very very few data points and very small datasets.

---

> > > ### Author Response · Authors · 2022-11-27
> > > **Authors Response**
> > >
> > > Thank you for your response.
> > >
> > > We respectfully disagree with some of the assertions and add some comments regarding the relative importance of the issues raised when considered in context.
> > >
> > > 1. **Tiny-NeRF**
> > >
> > > The results in Fig. 2 reflect the level of results obtained with the Tiny-NeRF architecture (not NeRF). The URL provided by the reviewer shows the ground truth test image on the top of the page. Then, it shows the results of the iterations. The results at the end of training are at the same level of detail as what we show for the baseline in Fig. 2 but it is smoother. In the original codebase, the number of points sampled along a ray is 64 while we used 32, due to the limitations of our GPU, which is the cause for the difference.
> > >
> > > A meta-discussion: Following the reviewer’s earlier request for  “a concrete application of the approach on a real-world setting that outperforms [all] existing baselines”, we are now state of the art in (1) speech separation, (2)  few-shot learning in NLP without prompts, and (3) TinyImageNet. These three results are obtained on very active benchmarks and we outperformed very recent baselines by a wide margin.
> > >
> > > Logic dictates that in these experiments the baseline network was at or below the state of the art and adding OCD surpassed the state of the art, so there can be no worry “that compared baselines are not fully trained”. Even if the reviewer is not convinced by the tiny-nerf experiments, how can these achievements be ignored?
> > >
> > > 2. **All layers**
> > >
> > > This is studied in our work, albeit, due to the required resources, only on the smaller datasets. To quote our paper:
> > >
> > > > On some datasets we check to what extent selecting a single layer limits our results, by performing the overfitting process on all of the model weights.
> > >
> > > and
> > >
> > > > Finally, we present an ablation that selects the layer with the second highest layer selection score, to evaluate the significance of the selection criterion.
> > >
> > > The conclusions of these ablations:
> > >
> > > > The ablation that selects an alternative layer results in performance that is similar to or slightly better than the base network. This is congruent with the small difference between fitting the selected layer and all layers, which may suggest that much of the benefit of overfitting occurs in the selected layer.
> > >
> > > In other words, overfitting the entire model is not much better than overfitting the selected layer. Therefore, the maximal gain one can obtain by applying OCD to all layers is expected to be small, not justifying the extra resources needed. The importance of the maximal varying layer in comparison to the other layers is verified by the lower success when running OCD on the alternative layer.
> > >
> > > A meta-discussion: if one wants to claim that the additional classification heads of GoogleLeNet are helpful, is it needed to show what happens when adding these heads after each block?
> > >
> > > One has to make use of the available resources, and in our case, we believe that preferring to show, for example, results for the same layer but in an ensemble (following A1Ss) is a choice that has led to (i) improvement in the state of the art results, and (ii) a clear demonstration of the unique power of our diffusion-model hypernetworks.
> > >
> > > Finally, if we were able to allocate the necessary resources to run OCD on all layers for the large-scale experiments, the likely outcomes would be either (i) no additional improvement over applying OCD to a single layer, (ii) a small improvement, (iii) a large improvement. There is also the unlikely outcome of (iv) reduced performance. However, none of these outcomes would invalidate the results (empirical or theoretical) that we currently have.
> > >
> > > 3. **Small scale experiments**
> > >
> > > While some experiments are done with small datasets and networks, this is not the case for all experiments.
> > >
> > > Speech separation: the architecture has 10.6M parameters and the training set has 20300 training samples.
> > >
> > > NLP: we employ an architecture of 110M parameters. When applying our OCD method, it surpasses the state-of-the-art architecture that has 3B parameters.
> > >
> > > Tiny ImageNet: this dataset was explicitly requested by reviewer A1Ss. It contains 100000 images of 200 classes, so it is not a small dataset. By applying OCD to the second-best base model, which has 87M parameters for the distilled version, we surpass the current state-of-the-art, which has 196M parameters. We tried working with the latter. However, we could not reproduce the baseline results (before applying OCD) so we moved to the former and had great success with it.
> > >
> > >
> > > A meta-discussion: when considering the impact of a body of experiments, in which all results are favorable, the process should not be min-pooling. A more fitting function is sum-pooling or max-pooling. We have made considerable efforts to satisfy the reviewer's request for additional state-of-the-art results and the existence of smaller-scale experiments alongside the requested large-scale ones cannot be held against us.

---

> > > > ### Comment · Reviewer_wwSB · 2022-11-27
> > > > **Reviewer Response**
> > > >
> > > > I thank the authors for their reply -- in terms of reviewing I personally believe I am trying my best effort to take the max-pool of the paper. In my opinion, the paper has very limited methodological contributions (it proposes to train a diffusion model on the weights of a network obtained by overfitting to each data point) and underlying architectural choices / models would be those I would also do if I were solve this problem (in fact I had tried a very similar idea around a year in the past, using a very similar experimental setup with very limited success). This underlying problem specification is also quite similar to setting on meta-learning/MAML (of which there are no-baselines that are compared to) -- where one can imagine that instead of learning "fast adaptation weights" a diffusion model is used to parameterize adaptation to the problem on hand.
> > > >
> > > > Thus given limited novelty in both problem formulation and methodology I think the crucial thing that matters (the max of the paper) lies in its empirical performance. Diffusion models are emerging new tool -- so is it true that they can truly help in set a setting? I think this is crucial portion of this paper, but the none of the results establish this in a convincing manner. Le-Net for instance is applied on MNIST, the tiny-NeRF results look substantially poorer than the ones on the website etc (if you look at the images near the bottom they are much smoother in nature and the PSNR is still increasing). Therefore, when examining the max of the paper, I believe it is still below acceptance threshold for the paper.
> > > >
> > > > I also personally believe that it is important to examine how we may adapt more than a single layer in a neural networks. For instance, MAML (which is baseline that is not compared to in the paper) is often applied to adapt networks at different modularities, from a single hidden layer to multiple hidden layers in a neural network and I think is something that should be necessarily studied in this paper .

---

> > > > > ### Author Response · Authors · 2022-11-27
> > > > > **Thank you for the quick reply!**
> > > > >
> > > > > We truly appreciate this ongoing discussion.
> > > > >
> > > > > **Empirical results**
> > > > >
> > > > > We currently disagree regarding the quality of the tiny-NeRF, which was discussed in length in our previous reply, and the exact source of the difference was explained. Let us also mention that Le-Net was developed for MNIST and is still used as a pedagogical example on this benchmark.
> > > > >
> > > > > More important to us is that we do not understand why our state-of-the-art results on three challenging benchmarks, as explicitly requested by the reviewer, are not acknowledged at all. By being state of the art, and assuming that the authors of the baselines did their best to obtain high results on the same benchmarks, the issue of perhaps undertraining the baselines becomes irrelevant.
> > > > >
> > > > > **MAML and novelty**
> > > > >
> > > > > There are multiple types of novelty, including conceptual, technical, and theoretical.
> > > > >
> > > > > In his latest response, the reviewer mentions for the first time MAML as a challenge for OCD's conceptual novelty. We indeed did not consider MAML as related until now. There are important differences between OCD and MAML: (1) MAML solves a different problem, (2) MAML is applied for multiple samples and even in the one-shot case has one sample per class, (3) MAML is applied at inference time to samples that are labeled, (4) MAML is not using a hypernetwork but adaptation during inference, and (5) MAML is not applied to samples from the training set in a single task setting.
> > > > >
> > > > > Certainly, MAML cannot be applied as a baseline, as is implied by the last response of the reviewer. MAML requires one to tune the network on the test sample, which cannot be done without the label.
> > > > >
> > > > > Meta comment: The type of “X is like Y except that” argument is very generic. Using the same logic, our paper lacks novelty due to any one-shot work out there (seems more relevant to us than MAML).
> > > > >
> > > > > Indeed, conceptual novelty is often subjective, especially in very active fields of research. What is not disputed in the review is that we present considerable technical novelty since we are the first diffusion-based hypernetwork and tackle the associated challenges for the first time, e.g., focusing on change in weights and not on the value of the weights, and separating the scale of the change from its direction. We would like to also point to the theoretical novelty we present by showing that diffusion models are suitable for hypernetworks under reasonable assumptions.
> > > > >
> > > > > Finally, we note that in the original review by wwSB, novelty was marked as one of the paper's strengths.

---

> > > ### Author Response · Authors · 2022-11-28
> > > **Update regarding tiny-NeRF**
> > >
> > > Reviewer wwSB noticed a discrepancy between the results we present for tiny-NeRF and the ones on the official git. As mentioned in our earlier reply, the difference stemmed from a smaller batch size due to the limits of our GPU. Following the review, we just ran the same set of experiments with the original batch size of 64 points per ray.
> > >
> > > We are pleased to report that we were able to replicate the results of the original git. The updated Figure 2 can be found here [imgur link] (https://imgur.com/a/3zcwzBl) and the results are indeed much smoother. As before, our method's output is closer in quality to that of overfitting on the test view than to the image obtained by the baseline method.
> > >
> > > The updated table is below. As can be seen, OCD is able to improve by at least half of the gap between overfitting on the test samples and the baseline method.
> > >
> > >
> > > | Method                  | Lego       | Hotdog     | Drums      |
> > > |-------------------------|------------|------------|------------|
> > > | Base Model              | 0.010$\pm$0.004 | 0.012$\pm$0.004 | 0.015$\pm$0.003 |
> > > | Overfitting on test     | 0.006$\pm$0.001 | 0.008$\pm$0.001 | 0.009$\pm$0.003 |
> > > | OCD no scaling ablation | 0.009$\pm$0.008 | 0.012$\pm$0.005 | 0.011$\pm$0.008 |
> > > | OCD(ours)               | 0.008$\pm$0.006 | 0.009$\pm$0.004 | 0.010$\pm$0.004 |
> > >
> > >
> > > We thank the reviewer once more for the helpful feedback. We believe that at this point, with the exception of the reviewer’s wish for more results with multiple layers, we were able to answer all of the raised concerns.

---

> ### Author Response · Authors · 2022-11-29
> **Following the post rebuttal review update by wwSB**
>
> While we appreciate the reviewer’s time and energy, we feel that the treatment of our work has not been entirely consistent with the expected standards.
>
> *Novelty:*
>
> Nov 5. The reviewer writes:
> > The problem studied by the paper is interesting and is new to my knowledge
>
> Nov 27. The reviewer mentions for the first time MAML as a related work and as a missing baseline. We post a reply in which we list five key fundamental differences between our work and MAML (the two contributions seem unrelated).
>
> Also Nov 27, the reviewer updates the review with post-rebuttal comments. and points to “Meta Learning via Learned Loss” (ML^3) as a related work. ML^3 learns a loss function given a task and labeled samples. Even if only unlabeled samples were used (which is not the case in the published ML^3 work), ML^3 produces a loss function and not network weights. Needless to say, there is no hypernetwork, no local learning, no diffusion model, no overfitting on single training samples, etc.
>
> To be clear: the authors respectfully assert that OCD, MAML, and ML^3 are as different as possible for three meta-learning methods.
>
> We would like to point out a recurring issue of neglect: reviewer wwSB did not acknowledge our response on MAML, nor did the reviewer refute our claims. It would have been courteous not to ignore it before venturing to claim that there is an additional supposedly related work we missed.
>
> *Experiments:*
>
> There is one experiment that was in dispute. On Nov 26, the reviewer wrote that our results on tiny NeRF are worse than the original git. We replied on Nov 27  explaining that this stems from the smaller batch size used to fit our GPU. This did not convince the reviewer, judging from the post-rebuttal edit of Nov 27. On Nov 28, we posted results on a batch size of 64, obtained on a bigger GPU. We hope that the reviewer would be able to consider these results and put this matter behind us.
>
> It is frustrating to us that the reviewer neglects the state-of-the-art results we have on multiple large datasets with large networks and repeats a false claim regarding “very limited data on very small architectures”. This is despite multiple official comments, including our reply from Nov 27, which points to three SOTA results on competitive benchmarks.
> > While some experiments are done with small datasets and networks, this is not the case for all experiments. [1] Speech separation: the architecture has 10.6M parameters and the training set has 20300 training samples. [2] NLP: we employ an architecture of 110M parameters. When applying our OCD method, it surpasses the state-of-the-art architecture that has 3B parameters. [3] Tiny ImageNet: this dataset was explicitly requested by reviewer A1Ss. It contains 100000 images of 200 classes, so it is not a small dataset. By applying OCD to the second-best base model, which has 87M parameters for the distilled version, we surpass the current state-of-the-art, which has 196M parameters. We tried working with the latter. However, we could not reproduce the baseline results (before applying OCD) so we moved to the former and had great success with it.

---

### Author Response · Authors · 2022-11-05
**notes on the first revision**

Following the reviews, we are uploading a revised version of the manuscript. Changes are marked in red.

1. A state-of-the-art experiment in NLP is added, improving upon very recent (August and September 2022) results in few-shot learning. This experiment also demonstrates the successful application of our OCD method to transformers.
2. We have added a paragraph to motivate local learning in general.
3. We added a sentence to make the data collection process clearer.
4. More data on the computational complexity of our method is added, making it clear that computational complexity is not an imitation of OCD.
5. Large parts of the bib file were replaced to address issues with the way arxiv papers were cited. We also corrected a few other references.

---

### Author Response · Authors · 2022-11-07
**Notes on the 2nd revision (thank you Reviewer A1Ss)**

Following the suggestion of Reviewer A1Ss, we have experimented with applying an ensemble of OCD networks. Since the OCD hypernetwork is a diffusion process, its output depends on a random initialization. Therefore, one can readily obtain multiple local networks for every input sample $x$.

As it turns out, in the NLP experiments, in which OCD improves SetFit (Tunstall et al, 2022) beyond the state of the art of T-few B3 (Liu et al, 2022), applying an ensemble leads to an additional improvement across all five benchmarks.

---

### Author Response · Authors · 2022-11-09
**Notes on the 3rd revision (following the request of reviewer Q7pB)**

Following the request of reviewer Q7pB, we are uploading a revised version of the manuscript. The new version contains an additional table at the end of the experiments section (Tab. 8), which lists the measured runtime of the experiments with and without the OCD method.

---

### Author Response · Authors · 2022-11-15
**Notes on the forth revision, adding an experiment requested by A1Ss**

Following the request of A1Ss, we present results on TinyImageNet. Since we were not able to replicate the baseline results of the SOTA network (Swan L/4), we choose to work with DeiT-B/16-D. As shown in Tab. 2 of the revised version, we were able to improve the results of DeiT from 87.3% (our run) to 90.8%. Even more exciting, using the suggestion of reviewer A1Ss to use an ensemble of OCD networks with random initialization of the diffusion process, we were able to obtain 92.0%, which is 4.7% better than the baseline network, and surpassing the current SOTA of 91.4%.

---

### Comment · Area_Chair_FVxR · 2022-11-15
**Discussion**

Dear reviewers,

Your response to the authors' rebuttal would be highly appreciated.

Kind regards,
Your AC

---

### Author Response · Authors · 2022-11-15
**We appreciate your feedback**

Dear reviewers,

Thank you for your extremely helpful feedback, which greatly improved our manuscript.

We would highly appreciate feedback on our responses. We respectfully ask that the discussion would focus on the contributions of our work. By this point in time, we were able to establish that:
1. We revisit the neglected topic of local learning with an entirely new perspective.
2. We present a novel type of hypernetwork, which is the first hypernetwork that is based on diffusion models.
3. We show theoretically that if the input domain can be generated by a diffusion model, then a diffusion model can be used for a hypernetwork implementing our local learning framework.
4. We improve the (very recent) state of the art in two competitive domains (speech separation and few-shot NLP without a prompt) and provide a diverse set of other smaller-scale experiments. Following a specific reviewer request, we are now also the SOTA on TinyImageNet.
5. Reviewer A1Ss noticed that due to the random initialization of the diffusion process,  our hypernetwork generates an ensemble for free. Following this insight, we show that such ensembles are extremely beneficial.
6. All requests for elucidation have been fully addressed.

We, therefore, believe that we have made the case that our work is entirely novel, stands on solid theoretical grounds, and is beneficial for real-world machine learning (SOTA in three competitive tasks from three completely different domains).

If any of these claims are disputed for any reason, there is still time to discuss them.

Thank you,

The authors

---

### Author Response · Authors · 2022-11-17
**Notes on the fifth revision**

Following a recent comment by reviewer Nu1A, we have rearranged the revised manuscript to fit the page limit.

---

### Author Response · Authors · 2022-12-11
**Authors’ response to the late-breaking comment by Q7pB following wwSB's discovery**

We thank Q7pB for the note published today. We find it a sign of strength to our novelty claims that after a month of severe scrutiny by reviewer ssWb, the most similar work that was found within the vast landscape of the deep learning literature are the two mentioned in Q7pB’s note. During this time, completely unrelated work such as MAML and ML^3 were held against us without being retracted. Similarly, the two new contributions are very different from our work. For example, these contributions do not overfit samples of the training set, do not train a hypernetwork for local learning, and do not show theoretically or practically how to build a hypernetwork based on diffusion models (our main novelty claims). Their challenge to our work's novelty is, therefore, quite limited.

The relevant aspect of these contributions is that they perform local learning (local learning is exactly what Q7pB describes as "finetuning given a specific input without seeing the corresponding ground-truth output"). Local learning is not a new concept, and, even in hindsight, there is no particular reason to compare our work with these two contributions.

The first one studies a different problem. As the name “Tent: Fully Test-Time *Adaptation* by Entropy Minimization” suggests, it is designed and tested for a single-sample domain adaptation (including robustness to corruption). There is no reason to believe that it would be a valid baseline for our experiments since Wang et al. do not claim to work in vanilla single-domain settings.

The second method (“Tailoring: encoding inductive biases by optimizing unsupervised objectives at prediction time” by Alet et al., 2021) is more general in its problem definition, and, like our method, it employs meta-learning to local-learning (in a completely different way that is related to adaptive instance normalization [Huang and Belongie, 2017]). The method is extremely different than ours and is based on applying unsupervised learning on a dataset that is created by augmenting the test sample.

Tailoring was tested on synthetic datasets with very specific structures, in a very specific unsupervised setting of CIFAR-10, and as a way to protect against adversarial samples. In the latter domain, Alet et al. openly admit that their method is not the state of the art.

Please allow us to wonder why, after being scrutinized by ssWb for using small-scale datasets (repeatedly ignoring, despite our plea for some acknowledgment, the state-of-the-art results we have on three large-scale very-competitive benchmarks), there is a demand that we compare with a paper that was not tested against the state of the art methods on large-scale competitive benchmarks.

To summarize: these methods, which were suggested as baselines by wwSB not to us but to the other reviewers, are very significant contributions to the field and are based on clever ideas. However, both are not more related than other methods and are probably not very effective local learning methods.

Despite the limited relevancy, to remove any doubt by the reviewers, we would like to humbly ask the reviewers to delay judgment by 24hr in order for us to compare with TENT (Tailoring did not publish source code). We think this is only fair given the late-breaking nature of the demand that we compare with these methods, the crucial weight that is given to these by at least one reviewer,  and the noncollegial manner in which these contributions were pointed to by wwSB outside the public forum of openreview.

---

### Author Response · Authors · 2022-12-12
**Update on the additional baselines requested at the last minute**

Yesterday, we have written on the limited relevancy of the two new baselines suggested to both our novelty claims and the validity of our state of the art experiments. Briefly – the two methods are largely unrelated to ours, except for being local learning methods, and these methods do not seem competitive enough.

As We mentioned, there is an implementation for TENT but not for “Tailoring”. We promised to provide the results for TENT within 24 hours, and are now delivering on our promise.

We focus our experiments on the vanilla classification experiments, which are similar in nature to those in the TENT paper by Wang et al. The three benchmarks include the TinyImageNet dataset, which was requested by the reviewers during our joint journey toward a better manuscript, and which is a well respected benchmark for which OCD (with an ensemble) is now the SOTA.

As shown in the table below, TENT (the official implementation) is not competitive with OCD. In addition to the default of 10 TENT iterations, we also provide results for 1 iteration, as shown by Wang et al., and for 30 iterations. For all of these benchmarks and baseline configurations, OCD is markedly better than TENT. TENT is able to improve upon the baseline on MNIST and on TinyImageNet. In the latter case by almost a full percent. However, OCD (even without an ensemble) is more than 2.5 percents higher in accuracy.

| Method / Accuracy %       | CIFAR10 (GoogleNet) | MNIST (LeNet5) | TinyImageNet (DeiT-B/16-D) |
|---------------------------|---------------------|----------------|----------------------------|
| Baseline                  | 92.85 \pm 0.40      | 99.2 \pm 0.1   | 87.29 \pm 0.15             |
| Baseline + Tent (1 iter)  | 92.79 \pm 0.40      | 99.4 \pm 0.1   | 88.15 \pm 0.20             |
| Baseline + Tent (10 iter) | 92.80 \pm 0.41       | 99.4 \pm 0.1   | 88.20 \pm 0.20             |
| Baseline + Tent (30 iter) | 92.82 \pm 0.40      | 99.3 \pm 0.1   | 88.20 \pm 0.18             |
| Baseline + OCD            | 93.68 \pm 0.38      | 99.7 \pm 0.1   | 90.80 \pm 0.21             |


Last note: as the discussion window nears its end we would like to thank the reviewers. We have been through a lot, including what we have experienced as challenging treatment by one of the reviewers, who ignored our results and kept coming back with additional criticism without ever stopping to acknowledge that the previous criticism was misplaced or fully addressed. On the flip side, our manuscript has greatly benefited from the feedback of all reviewers, regardless of their style, and we believe that in its current form it has surpassed all required standards of novelty, interest, clarity, and empirical success.

---

### Decision · Program_Chairs · 2023-01-20

**Decision:**

Reject

**Justification For Why Not Higher Score:**

The idea seems interesting, and the experimental results promising, but the presentation is below the bar for ICLR.

**Justification For Why Not Lower Score:**

N/A

**Metareview: Summary, Strengths And Weaknesses:**

Final Ratings: 3/6/5/5.
Confidence: 4/3/4/3.
Recommendation: Reject.

Let p(y|x;w) be a classification model, with 'x' the input and 'y' the label, with parameters 'w'. Let 'w_s' be the parameter if, starting from 'w', we would consequently overfit the model on a newly observed s=(x,y). Let p(w_s|x) be the distribution over these over-fitted parameters. The paper proposes to learn a model of p(w_s|x), and proposes to use a diffusion model for this task.

The reviews were mixed, even after rebuttals, mainly due to the lack of clear presentation. We had an AC/Reviewer discussion. See below. Reviewer Q7pB reduced his rated from an 8 to a 5 after the AC/Reviewer meeting.

The idea seems interesting, and the experimental results promising, but the presentation is below the bar for ICLR. I propose that the authors work to fix this issue and re-submit to another venue.

**Summary Of Ac-Reviewer Meeting:**

The reviews were mixed, even after discussions. We had an AC/Reviewer discussion. After the discussion, a '1' rating was bumped up to a '3', but the sentiment was overall still negative. The reviewers generally agreed that the idea is somewhat interesting, and that the experiments look somewhat promising. However, multiple reviewers found the discussion of related works lacking. Multiple reviewers indicated that they found the motivation unclear. More worryingly, the presentation was found unclear, and contained multiple errors. One reviewer, Q7pBW, was originally positive about the paper, but changed his score from an 8 to a 5, mostly because of missing comparisons to related work.

Due to the high variance in ratings, I read the paper myself. My conclusion is that the presentation needs more work, and seems wrong at points. Eq 1: Unclear where this MSE loss is used? And why MSE? In Section 3: saying H() doesn't make sense: it's not defined as a loss, but as a function from datapoint to parameters. The arguments given for the statement that original parameters w the overfitted parameters w_s are close, are not very convincing. It seems to me that the optima can be quite far from eachother. Section on 'Layer selection' method is not clearly written, and the method is not clearly motivated. And, how many samples of theta_L? The section on the 'diffusion process' is very unclearly written (and I have a strong background in diffusion models quite).